# Aerosol effects on convective storms under pseudo-global warming conditions: insights from case studies in Germany

Lina Lucas<sup>1</sup>, Christian Barthlott<sup>1</sup>, Corinna Hoose<sup>1</sup>, and Peter Knippertz<sup>1</sup>

<sup>1</sup>Institute of Meteorology and Climate Research Troposphere Research (IMKTRO), Karlsruhe Institute of Technology (KIT), Karlsruhe, Germany

Correspondence: Lina Lucas (lina.lucas@kit.edu)

**Abstract.** Germany is heading toward a future with warmer temperatures due to climate change, and potentially cleaner air from electrification and stricter emissions regulations. But how will these evolving environmental conditions affect severe convective storms? This study addresses this question by simulating three supercell events in high resolution using the ICOsahedral Non-hydrostatic (ICON) model. The events observed during the Swabian MOSES field campaigns in 2021 and 2023 are analysed using the pseudo-global warming approach to assess their evolution in a warmer climate. Aerosol effects were incorporated across all temperature scenarios using a double-moment microphysics scheme. The effects of acrosols on clouds and precipitation were considered using a two-moment microphysics scheme in four temperature rise scenarios, providing detailed insight insights into the underlying microphysical mechanisms. The results show The results indicate that higher temperatures generally enhance convection, leading to resulting in more intense convective cells, increased precipitation amounts, and more extreme rainfall and hail events. Additionally, warmer conditions increase the likelihood of supercell formation and more intense mesocyclones. An important finding is that hailstones grow larger under lower CCN-cloud condensation nuclei (CCN) concentrations, and the area affected by large hail expands by up to unit400%400%, indicating growing severity and reach of hail events. In some cases, precipitation increases exceed 7 % K<sup>-1</sup>, indicating super-Clausius-Clapeyron scaling and suggesting that additional dynamical and microphysical processes amplify rainfall beyond thermodynamic expectations. In addition, lower CCN concentrations are associated with a reduced cold-to-warm rain formation ratio and decreased precipitation efficiency. These aerosol-related effects appear largely independent of temperature, showing consistent patterns across all simulated warming scenarios. These findings indicate the intensity of severe weather events, such as convective storms and flash floods, may increase in a future climate.

#### 1 Introduction

Convective storms, particularly supercells, are significant weather phenomena known for their destructive potential. They can produce severe weather events such as heavy rainfall, strong winds, lightning, and large hail (Markowski and Richardson, 2010). During a three-day period around 23 June-For example, from June 22 to 24, 2021, multiple severe hailstorms struck southern Germany, France, Switzerland, Austria, and the Czech Republic, culminating-resulting in approximately EUR 4 billion of damage (Kunz et al., 2022a). On 23 June 2021, a supercell formed in southwestern Germany, resulting in hailstones

reaching maximum diameters of up to 4 cm and ground hail accumulations reaching depths of 30 cm, accompanied by heavy rainfall, and associated flooding .In (Kunz et al., 2022a). In Europe, the highest hail frequencies occur during summer along the slopes of high mountain ridges, such as the Alps, Pyrenees, and the Carpathians, aligning with observed lightning hotspots (Cui et al., 2025). However, maps of mesocyclonic storm tracks between April and September 2013 to 2017 show that there is no particular region prone to supercells in Germany (Wapler, 2021). In recent years, the frequency of weather-related hazards has increased, largely attributed to a warming climate (Hoeppe, 2016; Rädler et al., 2018; Púčik et al., 2019; Taszarek et al., 2019; Raupach et al., 2021). In the period 1950–2021, additive logistic models applied to the ERA5 reanalysis indicate significant increases of lightning and hail across most of Europe (Battaglioli et al., 2023). Especially Germany and the Alps experienced an increase in the annual occurrence of extreme weather events involving lightning, hail exceeding 2 cm in diameter, and wind speeds surpassing 25 ms<sup>-1</sup> (Rädler et al., 2018).

Recent modelling studies have explored how convective storms respond to rising temperatures using the pseudo-global warming (PGW) approach, which isolates thermodynamic effects by modifying initial and boundary conditions in high-resolution simulations. In this framework, large-scale circulation patterns are preserved, but the temperature profile in the initial and boundary conditions is uniformly increased, and humidity is adjusted according to the Clausius–Clapeyron relationship under constant relative humidity. This ensures that only thermodynamic effects are perturbed, while synoptic-scale dynamics remain unchanged. These studies confirm that warming can intensify deep convection and hail production, though the magnitude and spatial characteristics of these changes vary with model configuration and regional context (Feng et al., 2024; Lin et al., 2024; Tahara et al., 2025). As a result, considerable uncertainty remains regarding how convective hazards such as hailstorms or supercells will evolve in a future climate.

35

The Clausius–Clapeyron (CC) relationship provides a thermodynamic estimate that atmospheric moisture, and hence precipitation intensity, increases by approximately 6–7 % per degree of warming. However, a number of observational and modelling studies report larger increases, a phenomenon known as super-CC scaling. Such behaviour, where precipitation intensifies beyond 7 % K<sup>-1</sup>, has been documented in convective and orographic rainfall events (Lenderink and Van Meijgaard, 2008; Haerter and Berg, 2009; Chen et al., 2024). These deviations are thought to result from a combination of dynamic and microphysical processes, including convective organisation (Semie and Bony, 2020), stronger updraughts and latent heating (Lenderink et al., 2017), mesoscale circulations (Fowler et al., 2021), and aerosol–cloud interactions (Martinkova and Kysely, 2020). Large-scale (synoptic) forcing may also play a role. In some cases, stronger synoptic lift or enhanced moisture convergence under warming could intensify convection and contribute to super-CC responses (Fowler et al., 2021). However, the degree to which such effects contribute remains uncertain.

Alongside thermodynamic changes, aerosol concentrations, especially those of cloud condensation nuclei (CCN), play a crucial role in shaping convective storm behaviour (Tao et al., 2007; Barthlott et al., 2022b; Thomas et al., 2023). CCN influence cloud microphysics by affecting droplet formation, growth, and the phase transitions within clouds, which in turn modulate precipitation efficiency, storm dynamics, and hail formation (Fan et al., 2016)(Fan et al., 2016; Allen et al., 2020). Due to stricter emission regulations, anthropogenic aerosol concentrations are projected to decline in North America and Europe (Smith et al., 2011; Leibensperger et al., 2012; Genz et al., 2020), while economic growth in regions such as Asia may lead to increased CCN

emissions. In addition to anthropogenic sources, natural aerosols, such as dust, sea salt, volcanic emissions, and biomass burning, also contribute significantly to aerosol variability and may respond to climate drivers like temperature and wind changes (Chin et al., 2014; van Oldenborgh et al., 2021; Yli-Juuti et al., 2021). Moreover, aerosols remain a major source of uncertainty in climate projections, particularly in simulating extreme weather events (Watson-Parris and Smith, 2022)

Several recent studies have reported that warming alters convective environments, favouring more intense systems while suppressing weaker ones (Rasmussen et al., 2020; Huang et al., 2024; Mallinson et al., 2024; Yang et al., 2024). For instance, Mallinson et al. (2024) observed larger hailstones primarily in the cold season, while Trapp et al. (2019) reported increases throughout the year. Despite growing interest in this area, the future behaviour of extreme convective storms remains uncertain. In parallel, the forecasting skill of numerical weather prediction (NWP) models has improved considerably over recent decades, thanks to advances in computational power, data assimilation, and parametrisation schemes (Magnusson and Källén, 2013; Bauer et al., 2015; Buizza, 2019). However, predicting convective precipitation remains a challenge due to the complexity and non-linearity of the involved processes. Storm initiation and evolution are influenced by factors such as synoptic-scale flow, orography, and land surface heterogeneity (Kirshbaum et al., 2018). On a microphysical scale, aerosol-cloud interactions add further uncertainty. Acting as CCN, aerosols influence droplet size and precipitation processes. High CCN concentrations typically result in more numerous, smaller droplets, delaying precipitation onset but potentially intensifying storms through latent heat release at higher altitudes (Seifert et al., 2012). This process is described by the cold-phase convection invigoration theory, which suggests that enhanced freezing of supercooled droplets in polluted environments increases latent heat release in the upper parts of clouds, thereby strengthening updraughts and storm intensity (Rosenfeld et al., 2008). However, previous studies could not confirm this theory using the ICON model (Barthlott et al., 2022a), this theory remains highly debated: while some studies report stronger storms under polluted conditions, others find weaker or negligible effects depending on model setup and environmental factors (Altaratz et al., 2014a; Fan et al., 2018; Igel and van den Heever, 2021). Moreover, Barthlott et al. (2022a) found a systematic decrease in precipitation with increasing aerosol loading for cases with weak synoptic forcing, but nonsystematic responses for some of the strong synoptic forcing realisations. Therefore, this study investigates the response of convective storms to global warming and aerosol effects in Germany. Its uniqueness lies in the combined consideration of temperature changes and varying CCN concentrations. This allows for a more comprehensive understanding of how multiple drivers may shape future storm behaviour. The analysis focuses on convective parameters and precipitation characteristics, with particular attention to hail formation and supercell intensities. While particular attention is given to supercell-relevant metrics such as updraught helicity, the domain-mean evaluations also include other forms of convection (e.g., single cells and multicells) that occurred alongside the supercells in the chosen case studies.

To support this integrated analysis, the study employs high-resolution, convection-resolving-convection-permitting simulations with the ICOsahedral Non-hydrostatic (ICON) model at a 1 km grid spacing. This fine resolution is essential for explicitly resolving convection, enabling a more accurate representation of storm dynamics and temperature-induced effects. Additionally, the double-moment (2MOM) microphysics scheme of Seifert and Beheng (2006) is used to simulate aerosol effects on both liquid and mixed-phase clouds. By considering both temperature and aerosol influences, this approach provides new insights into the microphysical mechanisms shaping storm intensity, precipitation patterns, and supercell development under changing

**Figure 1. (a)** Orography of Central Europe (m above mean sea level) with the black box indicating the simulation domain with a 1 km grid, the blue box denoting the Germany evaluation domain, and the red box the Swabian MOSES field campaign domain. **(b)** Orography of the MOSES area, showing a zoomed-in view of the region marked by the red box in (a).

environmental conditions. The study focuses on real-case simulations of supercell events in Germany, including cases from the Swabian MOSES field campaigns conducted in the summers of 2021 and 2023 in southwestern Germany (Fig. 1).

The remainder of this paper is structured as follows: Section 2 describes the model setup, discusses cases and methodology,

Section 3 presents the results of the evaluation, and Section 4 summarises the key findings and conclusions.

#### 2 Method




#### 2.1 Model description

All simulations in this study are conducted using version 2.6.6 of the ICON model (Zängl et al., 2015), which employs a triangular gridstructurean icosahedral-triangular Arakawa C grid. The simulation domain covers Germany and parts of neighbouring countries (Fig. 1), with a horizontal grid spacing of 1 km. The vertical coordinates contain 100 terrain-following levels with a model top at a height of 22 km. ICON uses the smooth level vertical (SLEVE) coordinate, allowing a faster transition to a smoother vertical grid (Leuenberger et al., 2010)(Schär et al., 2002; Leuenberger et al., 2010). Furthermore, the model uses the 2MOM double-moment microphysics scheme of Seifert and Beheng (2006) to simulate aerosol effects on liquid and mixed-phase clouds. The scheme predicts the mass and number densities of six hydrometeor types: cloud droplets, raindrops, cloud ice, snow, graupel, and hail, and parametrises homogeneous and heterogeneous nucleation processes, including cloud condensation nuclei activation. Also, turbulence effects on droplet coalescence, raindrop breakup, and size-dependent collision efficiencies are taken into account. Ice nucleating particle (INP) concentrations were set to summertime conditions for immersion and deposition freezing in Germany (Hande et al., 2015). Further model settings are given in Table 1.

**Table 1.** Model configuration of the ICON simulations.

| Model aspect                 | Setting                                                                                      |
|------------------------------|----------------------------------------------------------------------------------------------|
| Initial and boundary data    | 7 km ICON-EU analyses, 3 h update                                                            |
| Heterogeneous ice nucleation | Based on typical mineral dust concentrations (Hande et al., 2015)                            |
| Homogeneous ice nucleation   | Following Kärcher and Lohmann (2002) and Kärcher et al. (2006)                               |
| CCN activation               | Calculated using pre-calculated activation ratios provided by Segal and Khain (2006)         |
| Deep & shallow convection    | Resolved                                                                                     |
| Land-surface model           | Multi-layer land-surface scheme TERRA (Heise et al., 2006)                                   |
| Turbulence parametrisation   | 1D based on the prognostic equation for the turbulent kinetic energy (Raschendorfer, $2001)$ |
| Radiation scheme             | ecRAD (Hogan and Bozzo, 2018), called every 12 min                                           |

## 2.2 Simulations overview

To investigate the impact of aerosols on convective storms in a warmer climate, each case is simulated for five different temperatures, with four CCN concentrations applied to each scenario.

## 1. *Temperature*



This study investigates the response of convective storms to a warming climate using a PGW approach (Brogli et al., 2023). For each day, five temperature scenarios are simulated, including one reference case. In the remaining four scenarios, the atmospheric temperature is uniformly increased throughout the entire column by increments of  $\Delta T = +1, +2, +3, +4$  K. To maintain consistency with the temperature adjustments, the specific humidity is recalculated using the Clausius-Clapeyron relation under the assumption of constant relative humidity. Additionally, to ensure physical consistency, the pressure at each model level is adjusted by numerically integrating the hydrostatic balance equation:

$$\frac{dp(z)}{dz} = -g \cdot \frac{p(z)}{R \cdot T(z)} \tag{1}$$

from the surface to the top of the atmosphere, using the reference simulation's surface pressure as the initial condition (Schär et al., 1996; Kröner et al., 2017). Hereby, g represents the gravitational acceleration, p(z) the pressure at height z, R the specific gas constant for dry air, and T(z) the temperature profile as a function of height. Furthermore, soil temperature , and soil surface temperature , and surface temperature are adjusted accordingly to match the imposed temperature increments are adjusted by applying the same temperature increments as imposed on the atmosphere.

| Label | Concentration         | CCN Regime           |  |  |
|-------|-----------------------|----------------------|--|--|
| C1    | $100{\rm cm}^{-3}$    | Maritime             |  |  |
| C2    | $250\mathrm{cm}^{-3}$ | Intermediate         |  |  |
| C3    | $1700{\rm cm}^{-3}$   | Continental          |  |  |
| C4    | $3200{\rm cm}^{-3}$   | Continental Polluted |  |  |

Table 2. Classification of cloud condensation nuclei (CCN) concentrations and their corresponding labels (Segal and Khain, 2006).

### 2. CCN concentration



The CCN concentrations used in the simulations are listed in Table 2. C3 is chosen as the reference concentration, as it closely represents present-day CCN levels observed over Germany (Hande et al., 2015). The activation of CCN from aerosol particles is calculated using look-up tables with pre-calculated activation ratios provided by Segal and Khain (2006). In that scheme, four different CCN concentrations ranging from low to very high concentrations can be chosen (Table 2). The continental CCN concentration (C3) is chosen as the reference concentration, as this aerosol assumption represents typical conditions of central Europe (Hande et al., 2016; Costa-Surós et al., 2020).

In addition, the applied PGW method, which assumes vertically uniform warming while preserving relative humidity, does not capture potential changes in the lapse rate (e.g. Brogli et al., 2021) or modifications in large-scale weather systems that may result from a weakened polar jet stream. Likewise, humidity gradients that could emerge over extended simulation periods are not represented. Nevertheless, our findings, generated with an advanced double-moment microphysics scheme that explicitly resolves hail, offer important insights into the influence of aerosols on mixed-phase clouds under various warming conditions.

#### 140 2.3 Analysed cases

This study examines three supercell events from the Swabian MOSES field campaigns conducted in 2021 (Kunz et al., 2022b) and 2023 (Handwerker et al., 2025, submitted) (Handwerker et al., 2025). The 2021 campaign aimed to investigate convective storms associated with heavy rain and hail and large-scale heatwaves associated with droughts, while in 2023, the emphasis was on convective storms associated with heavy rain and hail, localised flooding, and pollutant inputs to water bodies. It focused on capturing entire events from their onset through development to their impact. Both campaigns were conducted in southwestern Germany, including the Neckar Valley and Swabian Alb (Fig. 1). This area was selected because hailstorms most frequently occur south of Stuttgart and over the Swabian Alb (Kunz and Puskeiler, 2010; Puskeiler et al., 2016). To evaluate the accuracy of the ICON model simulations, results are compared to radar-derived precipitation data from the RADOLAN (Radar Online Adjustment) algorithm provided by the German Weather Service (DWD). No systematic comparison with observations and no evaluation in this direction are carried out, as this is not the goal of the present study. The RADOLAN data are only used to ensure that the simulations reproduce the general precipitation pattern of the selected day. The study is designed as a sensitivity study, focusing on how the model responds to changes in the prescribed parameters. By analysing these simulations across

**Figure 2.** (a, c, e) Global Forecast System (GFS) analysis at 12:00 UTC showing 500 hPa geopotential height (gpdm; shading), sea-level pressure (hPa, red contours), and 500 hPa wind barbs for case 1 (23 June 2021), case 2 (28 June 2021), and case 3 (22 June 2023). (b, d, f) Accumulated precipitation over 24 hours across the entire Germany domain from the reference simulation using continental CCN concentrations (mm). The black box indicates the MOSES domain as in Fig. 1.

different atmospheric conditions, this study captures a diverse range of convective storm characteristics. Each case represents a distinct convective scenario, highlighting variations in convective cell behaviour (Fig. 2).

## 155 2.3.1 Case 1: 23 June 2021

On 23 June 2021, a weakening trough stretched from southern Scandinavia to southern Spain (Fig. 2a). During the day, a shortwave trough moved along its eastern side and reached Germany in the early afternoon. Deep convection occurred with an

increase in convective available potential energy (CAPE) in the middle layer over southwest Germany to around  $950 \,\mathrm{J\,kg^{-1}}$  by midday. Moderately strong winds from a south-westerly direction prevailed in the middle troposphere over Germany. At the surface, a large area of high pressure extended over the North Atlantic, across western Europe, and to northern Germany. A stationary front, a remnant of a multi-core area of low pressure from the previous days, which was now connected to a surface low over Finland, lay over southern Germany and persisted in the following days (Barthlott et al., 2024). The deep layer shear (DLS) reached  $18 \,\mathrm{m\,s^{-1}}$  on average in the MOSES domain, indicating suitable conditions for supercell formation.

## 2.3.2 Case 2: 28 June 2021


On 28 June 2021, a trough was located over the west coast of France and covered the area from Great Britain to Spain (Fig. 2c). Over Germany, a slightly strengthening ridge was present, while a short-wave trough passed over southwest Germany in the afternoon. As a result, Germany was affected by moderately strong winds coming from the southwest. A high-pressure field formed over the Atlantic, and pressure values around 580 hPa were reached in the middle troposphere over Central Europe. Averaged over the Germany evaluation domain, DLS reached values of 20 m s<sup>-1</sup> which also reflects the potential of supercell formation. Intense hailstorms also occurred over Switzerland on that day: Kopp et al. (2023) provide a synoptic discussion, hail-storm tracks, and radar data, also for southwestern Germany, whereas Brennan et al. (2025) present a comprehensive model-based case study to unravel the complex processes involved in the genesis, intensification, and dissipation of this impactful weather event.

#### 2.3.3 Case 3: 22 June 2023

On 22 June 2023, the synoptic setup featured a prominent trough over the North Atlantic and France and a ridge over southeastern Europe (Fig. 2e). This pattern was associated with a low-pressure system to the northwest and a high-pressure system to the southeast. The resulting pressure gradient led to strong southwesterly flow across Central Europe, advecting warm and moist air and supporting large-scale ascent that contributed to the initiation of convection. As on the other days, suitable conditions for supercell formation were simulated here with DLS values of 19 m s<sup>-1</sup> in the MOSES domain.

#### 180 **2.4 Evaluation techniques**

The following analyses apply different techniques to quantify key aspects of convective storms, including cold pool dynamics, hailstone size distributions, and the occurrence and evolution of convective cells. It is important to emphasise that, although the selected case days were chosen because they featured supercell events, the diagnostics are not limited to supercells. Many of the evaluations, such as domain-mean precipitation, cold pools, or convective cell number or lifetime, inherently capture the behaviour of all convective cells within the model domain. This means that the results reflect the combined response of single cells, multicells, and supercells that coexist during the simulations. The methods outlined in the following subsections, therefore, provide insight into the overall convective response, while still allowing for the interpretation of processes particularly relevant to supercells.

## 2.4.1 Cold pools

Cold pools offer additional insight into convective intensity. They consist of volumes of negatively buoyant air that originate from evaporative cooling in precipitating downdraughts (Hirt et al., 2020). They are characterised by negative temperature anomalies, which can be derived using the density potential temperature defined by Hirt et al. (2020) as:

$$\theta_{\rho} = \theta (1 + 0.608r_v - r_w - r_i - r_r - r_s - r_q - r_h) \tag{2}$$

where  $\theta$  is the potential temperature,  $r_v$  is the water vapour mixing ratio, and  $r_w$ ,  $r_i$ ,  $r_r$ ,  $r_s$ ,  $r_g$ , and  $r_g$  are the mixing ratios of cloud water, cloud ice, rain, snow, graupel, and hail, respectively. The local perturbation of density potential temperature  $\theta'_{\rho,0}$  is calculated relative to a spatiotemporal moving average  $\bar{\theta}^m_{\rho}$ , using a horizontal window of 166 grid points ( $\approx$  166 km) and a temporal window of 8 hr as in Hirt et al. (2020). They also conducted sensitivity analyses with different spatial filtering scales and with/without temporal filtering, which did not show a strong influence on the qualitative behaviour of the results. For that reason, the same values are used in our study. This combination of spatial and temporal filtering enables the identification of large cold pools while background  $\theta_\rho$  gradients at the coast are still sufficiently resolved to detect cold pools there (Hirt et al., 2020). To isolate cold pool structures, the domain-mean perturbation is subtracted:  $\theta'_{\rho} = \theta'_{\rho,0} - \bar{\theta}'_{\rho,0}$ . Cold pools are identified by applying thresholds: only perturbations below -2 K, associated with precipitation rates exceeding 10 mm hr<sup>-1</sup>, and with horizontal extents greater than 6 km are considered. The remaining cold pool perturbations are used to compute a density distribution function for each temperature and CCN concentration scenario.

## 205 **2.4.2** Hail sizes




To evaluate hailstone sizes in the model, hailstone size distributions are calculated using a generalised Gamma distribution:

$$f(x) = Ax^{\nu} \exp(-\lambda x^{\mu}), \tag{3}$$

whereby  $\nu$  represents the shape parameter,  $\mu$  the dispersion parameter and x is the particle mass. A and  $\lambda$  are calculated from the predicted mass and number densities. Different combinations of  $\nu$  and  $\mu$  with values between 0 and 1 lead to different distributions. If  $\mu=1$ , the function reduces to a classical  $\Gamma$ -distribution, for  $\nu=\mu=1$  the result is the Weibull distribution, and for  $\nu=\mu=0$  an exponential distribution is calculated (Seifert and Beheng, 2006). For this evaluation,  $\nu=1$  and  $\mu=\frac{1}{3}$  are used, which are the parameters for hail in this model. This function is rewritten as a function of diameter D, using  $x=\frac{\pi}{6}\rho D^3$  which is true for spherical particles (Khain et al., 2015):

$$f(D) = N_0' D^{\nu'} \exp(-\lambda' D^{\mu'}),$$
 (4)

with  $N_0' = 3N_0(\frac{\pi}{6}\rho)^{\nu+1}$ ,  $\nu' = 3\nu + 2$ ,  $\lambda' = \lambda(\frac{\pi}{6}\rho)^{\mu}$ , and  $\mu' = 3\mu$ .  $N_0$  represents the intercept parameter and  $\rho$  the bulk hydrometeor density. This formulation is used to derive hailstone size distributions for the subsequent analysis.

#### 2.4.3 Tracking of convective cells

The occurrence of convective cells is analysed using version 1.5.3 of the Python package Tobac (Tracking and Object-Based Analysis of Clouds (Heikenfeld et al., 2019; Sokolowsky et al., 2024). Tobac is a software to identify, track, and analyse clouds and other meteorological phenomena such as updraughts, precipitation, and radiation. The tool uses algorithms to identify features and link them into consistent trajectories.

Tobac links features between consecutive time steps using a nearest-neighbour and overlap-based approach. When a feature splits into multiple successors, the trajectory continues along the largest successor while the others are recorded as new tracks. In the case of mergers, the trajectory of the largest predecessor is continued, and the additional features are terminated. In this study, the number of convective cells refers to the number of unique tracks initiated during the simulation period.

In this study, updraughts of cells at high temporal resolution (5 min) between 06:00 UTC and 24:00 UTC are tracked. For the updraught tracking, first, the data of the maximum updraught between 3 and 8 km height and the total condensate mixing ratio, which means the total amount of liquid and frozen water per mass of dry air, is calculated. Afterwards, several thresholds (3, 5,  $10 \,\mathrm{m\,s^{-1}}$ ) are used to track updraughts to better resolve overlapping cells. In addition, only updraughts with a minimum area of 10 grid cells are used as features, and a maximum distance of 25 grid cells is selected when linking features. As a segmentation threshold, a value of  $0.5^{-3} \,\mathrm{kg\,kg^{-1}}$  is chosen as the condensate mixing ratio to identify the cloud volumes corresponding to the individual identified updraughts (Heikenfeld et al., 2019). To track only longer-lived cells, only tracks with a minimum lifetime of 30 min are used for the evaluation. The tracking approach, therefore, captures all convective cells within the domain, not only supercells.

## 3 Results







This ehapter-section presents the results of ICON simulations evaluating the sensitivity of convective storms to atmospheric warming and changes in CCN concentration. The analysis focuses on both thermodynamic and microphysical aspects of storm behaviour. ChapterSection 3.1 examines the intensity of convective cells, focusing on CAPE, CINconvective inhibition (CIN), and updraught helicity as indicators of supercell intensity. It also analyses how changes in temperature and CCN concentrations affect up- and downdraught velocities and cold pool intensities, providing insight into the resulting storm dynamics. ChapterSection 3.2 investigates precipitation responses, including changes in total and extreme rainfall and hail, deviations from Clausius—Clapeyron scaling, and the evolution of hailstone size distributions under future climate conditions. ChapterSection 3.3 analyses the frequency and lifetime of convective cells using object-based tracking methods. Finally, ChapterSection 3.4 explores aerosol—cloud interactions, detailing how CCN concentrations influence hydrometeor content, microphysical process rates, the cold-to-warm rain formation ratio, and precipitation efficiency.

#### 3.1 Intensity of convective cells

In all three cases, CAPE and convective inhibition (CIN) CIN exhibit an increasing trend with rising temperatures (not shown).

The. The maximum values of CAPE and the minimum values of CIN, taken from the mean daily time series, are summarised

| Temperatures              | CAPE   |        |        | CIN    |        |        |
|---------------------------|--------|--------|--------|--------|--------|--------|
|                           | Case 1 | Case 2 | Case 3 | Case 1 | Case 2 | Case 3 |
| Reference                 | 272    | 408    | 861    | 3.9    | 18.5   | 43.9   |
| $\Delta T = +1 \text{ K}$ | 347    | 516    | 984    | 4.5    | 20.0   | 44.3   |
| $\Delta T = +2 \text{ K}$ | 412    | 646    | 1121   | 5.9    | 22.0   | 46.7   |
| $\Delta T = +3 \text{ K}$ | 497    | 776    | 1258   | 7.5    | 25.0   | 46.8   |
| $\Delta T = +4 \text{ K}$ | 574    | 922    | 1355   | 9.7    | 27.6   | 47.9   |

**Table 3.** Maximum values of the time series of domain-mean CAPE and minimum values of CIN, shown for all three case studies and all temperature simulations.

in Table 3. The rate of increase is relatively uniform across the different simulated temperatures, indicating that the sensitivity of CAPE and CIN to temperature is largely independent of the magnitude of warming, particularly during the time of day when convection is most active. However, the absolute increase in CAPE varies between approximately among cases between 65–85 J kg<sup>-1</sup> across the cases, on case 1, 108–146 J kg<sup>-1</sup> on case 2, and 97–137 J kg<sup>-1</sup> on case 3. CIN exhibits only a modest enhancement, typically on the order of a few J kg<sup>-1</sup>. While maintaining the relative humidity, the dew point increases less than the temperature, leading to a higher lifting condensation level and a higher level of free convection. The increase in CAPE can be explained by the steeper moist adiabats for increased temperatures with higher equilibrium levels. Similar findings have been reported in previous studies (Rasmussen et al., 2020; Huang et al., 2024; Mallinson et al., 2024; Yang et al., 2024), which also observed increased CAPE and CIN in PGW experiments. These results suggest a more favourable environment for stronger convective systems due to enhanced CAPE while simultaneously creating convective systems if the triggering is sufficiently strong to overcome CIN and less favourable conditions for weaker trigger mechanisms due to the rise in CIN.

Since the cases involve supercell events, an additional parameter is used to quantify the intensity of such storms. Supercells are characterised by a rotating updraught known as a mesocyclone, which plays a crucial role in their organisation, longevity, and severe weather potential (Markowski and Richardson, 2010). A widely used parameter for identifying such rotating updraughts is the updraught helicity (UH), which quantifies the strength of vertically integrated rotation within a storm. UH can be calculated by integrating the product of the vertical velocity w and the vorticity  $\zeta$ :

$$UH = \int w \cdot \zeta \, dz \tag{5}$$

In this study, the product is integrated from  $2100 \,\mathrm{m}$  to  $5000 \,\mathrm{m}$  AGL, focusing on storm rotation in the lower to middle troposphere (Kain et al., 2008). UH provides a useful way to detect and quantify rotating convective storms, as higher values indicate stronger updraught rotation. By plotting-mapping UH, the presence and intensity of mesocyclones can be visualised (Weisman and Rotunno, 2000; Thompson et al., 2007). To identify potential supercell regions, a threshold of  $75 \,\mathrm{m}^2 \,\mathrm{s}^{-2}$  is applied, following the methodology of Ashley et al. (2023). To analyse helicity changes across as in Ashley et al. (2023). Although the threshold in Ashley et al. (2023) was established on  $4 \,\mathrm{km}$  resolution data, the same value is used here despite the higher

**Figure 3.** Mean helicity over the whole Germany domain of values larger than 75 m<sup>2</sup> s<sup>-2</sup> for (**a**) case 1 from 12:00 to 20:00 UTC, (**b**) case 2 from 12:00 to 20:00 UTC, and for (**c**) case 3 from 10:00 to 20:00 UTC. Number of grid points with helicity values larger than 75 m<sup>2</sup> s<sup>-2</sup> (**d**) case 1 from 12:00 to 20:00 UTC, (**e**) case 2 from 12:00 to 20:00 UTC, and for (**f**) case 3 from 10:00 to 20:00 UTC.

resolution of  $1\,\mathrm{km}$ , since the focus is not on the exact threshold itself but on the relative changes. Consistently, an additional analysis of the 99.95th percentile, similar to the approach of Wang et al. (2022), yielded values close to  $75\,\mathrm{m}^2\,\mathrm{s}^{-2}$ , supporting the choice of this threshold. Supercell dynamics are also strongly controlled by vertical wind shear. In the PGW framework applied here, the wind profiles are not systematically modified, so shear conditions remain close to those of the reference synoptic environments. Thus, the changes in helicity discussed below primarily reflect thermodynamic influences, while potential shear-related modifications are not represented.

Helicity changes across the different temperature simulations , the average helicity values exceeding are calculated, along with are assessed by computing the mean values above this threshold and the number of grid cells meeting this threshold that exceed it (Fig. 3). The analysis focuses on time frames associated with potential supercell development and the strongest convective activity: between 12:00 and 20:00 UTC for cases 1 and 2 and between 10:00 and 20:00 UTC for case 3. Across all three cases and CCN concentrations, higher temperatures lead to an increase in both the mean helicity values and the number of grid cells exceeding  $75 \, \text{m}^2 \, \text{s}^{-2}$ . These findings indicate that a warmer future may be associated with an increased likelihood of supercell formation, greater supercell intensity, or a combination of both. However, it is also found that mean helicity values plateau and, in some instances, decrease for the highest simulated temperatures.




While CCN concentrations do not show a uniform impact on helicity values across all simulations, distinct variations are observed in some cases between different CCN scenarios. In contrast, temperature shows a consistent and stronger influence, with helicity values increasing systematically under warmer conditions. Overall, the impact of temperature on helicity is more pronounced than that of CCN concentration.

**Figure 4.** Frequency difference of vertical velocity as a function of height and wind magnitude for different warming scenarios with respect to the reference temperature evaluated over the MOSES domain and the full 24-h simulation period. All panels represent continental CCN concentrations (C3).

**Figure 5.** Frequency difference of vertical velocity as a function of height and wind magnitude for different CCN concentrations with respect to the reference CCN concentrations (C3) evaluated over the MOSES domain and the full 24-h simulation period. All panels represent the reference temperature without global warming.

The influence of atmospheric warming on convective cell dynamics is illustrated by frequency differences of vertical winds as a function of height and magnitude in Fig. 4 for the 28 June 2021 case. case 2. As temperatures rise, both updraught and downdraught velocities intensify in the upper troposphere, indicating enhanced convective activity under warmer conditions. Updraughts in the range of 0–40 m s<sup>-1</sup> increase by approximately 50–100%, while the most intense updraughts show increases exceeding 300% in the +4 K simulation. A similar pattern is observed for downdraughts: lower intensities increase by 50–100%, with the strongest downdraughts (10–20 m s<sup>-1</sup>) increasing by more than 300% under the same warming scenario. This general intensification of vertical motion in the upper troposphere is consistently observed across all three analysed cases. However, it is primarily the moderate up- and downdraughts that show this robust increase in all cases, while the most extreme velocities exhibit more case-to-case variability. In contrast, changes in CCN concentration influence the frequency and distribution of vertical velocities. For higher CCN concentrations, the occurrence of strong updraughts (20–30 m s<sup>-1</sup>) in the upper troposphere decreases (not shownFig. 5). While downdraught intensity also weakens in the upper troposphere, a slight


**Figure 6.** Density distributions of cold pool temperature anomalies for different CCN concentrations, calculated between 06:00 UTC and 24:00 UTC within the MOSES domain for case 2. The domain is defined in Fig. 1.

increase is observed in the middle to upper troposphere. Near the surface, downdraught strength again diminishes. This vertical structure could be linked to enhanced precipitation loading aloft in a low CCN environment, which contributes to stronger downdraughts at mid-levels but less pronounced impacts near the surface.

To further investigate this behaviour the impact of CCN concentration on convective dynamics, cold pools are analysed. Cold pools offer additional insight into convective intensity. They consist of volumes of negatively buoyant air that originate from as an additional measure of storm intensity. Cold pools, which form through evaporative cooling in precipitating downdraughts(Hirt et al., 2020). They are characterised by negative temperature anomalies, which can be derived using the density potential temperature defined by Hirt et al. (2020) as:

$$\theta_{\rho} = \theta(1 + 0.608r_v - r_w - r_i - r_r - r_s - r_g - r_h)$$


where  $\theta$  is the potential temperature,  $r_v$  is the water vapour mixing ratio, and  $r_w$ ,  $r_t$ ,  $r_r$ ,  $r_s$ ,  $r_g$ , and  $r_g$  are the mixing ratios of cloud water, cloud ice, rain, snow, graupel, and hail, respectively. The local perturbation of density potential temperature  $\theta_{\rho,0}^t$  is calculated relative to a spatiotemporal moving average  $\bar{\theta}_{\rho}^m$ , using a horizontal window of 166 grid points and a temporal window of 8 hours. To isolate cold pool structures, the domain-mean perturbation is subtracted:  $\theta_{\rho}^t = \theta_{\rho,0}^t - \bar{\theta}_{\rho,0}^t$ . Cold pools are identified by applying thresholds: only perturbations below, associated with precipitation rates exceeding  $^{-1}$ , and with horizontal extents greater than are considered. The remaining cold pool perturbations are used to compute a density distribution function for each temperature and CCN concentration scenario. Varying CCN concentrations have a systematic influence, as illustrated in Fig. 6influence storm organisation and surface outflows. The methods used to identify and quantify cold pools in this study are described in Section 2.4.1. Figure 6 illustrates how varying CCN concentrations affect the strength and distribution of cold pool anomalies. The highest CCN case (C4) exhibits a pronounced peak around  $^{-5}$ K with a narrow distribution and a short tail, indicating frequent but moderate cold pool anomalies. As CCN concentrations decrease, the distributions broaden and develop longer tails, indicating greater variability and a higher occurrence of stronger cold anomalies.

**Figure 7.** Percentage change of the daily amount (24-h accumulated) of (a) rain and (d) hail, the change in the 95th percentile of (b) rain and (e) hail, and the change in the area affected by daily totals of (c) rain > 45 mm and (f) hail > 10 mm (the 95th percentile) for changing atmospheric temperatures on all three simulated days (different line colours) for a continental CCN concentration (C3) and in the MOSES domain for Cases 1 and 2 and the Germany domain for case 3. The domains are defined in Fig. 1.

This enhanced cold pool intensity at lower CCN concentrations is also reflected in the downdraughts: increased downdraught strength is observed in the lower troposphere under low CCN conditions (not shown). This behaviour is consistent with the known mechanism in which enhanced subcloud evaporative cooling leads to stronger cold pool formation and, consequently, more intense downdraughts. The increased evaporation under low CCN conditions is further promoted by the formation of larger raindrops, which evaporate more efficiently as they fall through the subcloud layer, enhancing cooling.

# 3.2 Precipitation




# 3.3 Precipitation

To evaluate changes in precipitation, the behaviour of rain and hail is analysed separately for all days (Fig. 7). First, the total amount of rain and hail is examined. Rain exhibits an increasing trend across all days, with increases ranging from approximately 15% to 70% in the +4 K simulation. Hail also increases with rising temperatures, exhibiting a greater relative change compared to rain. On 23 June 2021 and 28 June 2021In cases 1 and 2, hail increases by around 30%. However, on 22 June 2023in case 3, hail increases by nearly 200%, suggesting that synoptic-scale forcing likely played a role. Synoptic-scale environments significantly influence the response of hailstorms to climate warming, with frontal systems exhibiting a more

substantial increase in large hail occurrences compared to other synoptic setups (Fan et al., 2022). In the present experiments, the imposed warming does not modify the large-scale dynamics, so the prevailing synoptic situation of each analysed day remains unchanged. In a warmer climate, however, storm tracks and jet streams are projected to shift poleward, and changes in cyclone and frontal characteristics are expected (Harvey et al., 2020; Priestley and Catto, 2022). Such changes would alter the frequency and structure of synoptic environments conducive to severe convection and hail, further underlining the importance of the background synoptic forcing for interpreting the results of this sensitivity study.

To assess changes in extreme precipitation events, the 95th percentile of rain and hail amounts is analysed. For rain, the 95th percentile consistently increases with temperature, indicating that even heavier rainfall events will occur, heightening the risk of flash floods. On 28 June 2021 and 22 June 2023In cases 2 and 3, extreme rainfall amounts increase by approximately 50% in a +4 K warming scenario. A similar trend is observed for hail, with a substantial increase in the 95th percentile. On 22 June 2023In case 3, extreme hail amounts rise by over 200%, indicating a strongly increased potential for severe hailstorms and elevated damage risks in a warmer climate.

Next, the number of grid cells experiencing extreme rainfall and hailfall (95th percentile) is evaluated. Again, a notable increase is observed. On 22 June 2023In case 3, a +4 K temperature increase results in a 400 % expansion of the area impacted by large hail amounts. This corresponds to an increase from 369 km² to 3,995 km², representing a dramatic growth from an area comparable to the size of Munich to one comparable to the island of Mallorca. Even the smallest increase, observed on 23 June 2021in case 1, amounts to 100 %, representing a doubling of the impacted region. The effects of CCN concentrations on precipitation will be discussed later in Chaptersection 3.4.

#### 3.2.1 Hail sizes



Since hail appears to undergo the most significant changes, and because Because hail damage assessments are primarily based on hailstone sizes (Kim et al., 2023; Schmid et al., 2024), the influences of higher temperatures and changes in CCN concentrations on hailstone sizes are investigated. The hailstone size distributions are calculated using a generalised Gamma distribution:

$$f(x) = Ax^{\nu} \exp(-\lambda x^{\mu}),$$

whereby  $\nu$  represents the shape parameter,  $\mu$  the dispersion parameter and x is the particle mass. A and  $\lambda$  are calculated from the predicted mass and number densities. Different combinations of  $\nu$  and  $\mu$  with values between 0 and 1 lead to different distributions. If  $\mu=1$ , the function reduces to a classical  $\Gamma$ -distribution, for  $\nu=\mu=1$  the result is the Weibull distribution and for  $\nu=\mu=0$  an exponential distribution is calculated (Seifert and Beheng, 2006). For this evaluation,  $\nu=1$  and  $\mu=\frac{1}{3}$  are used, which are the parameters for hail in this model. This function is rewritten as a function of diameter D, using  $x=\frac{\pi}{6}\rho D^3$  which is true for spherical particles (Khain et al., 2015):

$$f(D) = N_0' D^{\nu'} \exp(-\lambda' D^{\mu'}),$$

with  $N_0' = 3N_0(\frac{\pi}{6}\rho)^{\nu+1}$ ,  $\nu' = 3\nu + 2$ ,  $\lambda' = \lambda(\frac{\pi}{6}\rho)^{\mu}$ , and  $\mu' = 3\mu$ .  $N_0$  represents the intercept parameter and  $\rho$  the bulk hydrometeor density this section focuses on how hailstone sizes respond to variations in temperature and CCN concentration. The resulting mean hail size distributions on at the surface for 23 June 2021 case 1 are presented in Fig. ??8a. To assess 370 how hail sizes respond to changes in temperature and CCN concentration, the maxima of the distributions are analysed as indicators of the dominant hailstone size (Fig. 8b-d). The results reveal a systematic increase in hailstone size with increasing temperatures across all CCN concentrations. This occurs despite an elevation of the freezing level by approximately per degree Celsius, which would typically enhance hail melting and reduce final hail size. Furthermore, lower CCN concentrations are also associated with larger hail sizes. Temperature and CCN concentration exert similarly strong influences. In addition, the 375 simultaneous rise of the freezing level promotes melting of smaller hailstones, while the larger hailstones produced in stronger updraughts are much less affected due to their smaller surface area relative to their volume. As a result, the surface hail size distribution shifts toward fewer small hailstones but a greater occurrence of large ones, consistent with findings in the current literature (Raupach et al., 2021). Overall, variations in CCN concentration appear to have an impact on hail size of similar magnitude to that of temperature changes. Given that both future warming and decreasing CCN concentrations are expected, 380 these factors are likely to contribute to the formation of larger hailstones under future atmospheric conditions.

To evaluate the impact of temperature and CCN concentration on the largest hail sizes, the hail size distribution is extended beyond the model's initial output up to the point where the distribution curve returns to a relative frequency of 0.0025. In the model, the particle mass x is constrained by a maximum value  $x_{\rm max}$ , which limits the simulated hail size distribution (Fig. ??8a) and excludes larger hailstones hailstones larger than 2.3 cm. However, when comparing the extended size distribution curve for larger x with the curve of the model's output, both follow the same trend, indicating that the distribution can be reliably extended to account for larger hailstones. Therefore, the hail sizes corresponding to a distribution value of 0.0025 are determined from the extended hail size distribution curve and analysed using the same approach as for the dominant hailstone sizes. The results reveal an almost identical pattern for the changes in dominant and maximum hailstone sizes (not shown), confirming that not only do medium-sized hailstones grow larger in a warmer climate, but the large hailstones also experience significant strong growth. These findings are particularly concerning given the high damage potential associated with large hailstones.

#### 3.2.2 Clausius-Clapeyron scaling





To explain the observed increase in precipitation, Clausius–Clapeyron (CC) scaling provides an essential theoretical framework. Although uncertainties persist, it is widely accepted that rising temperatures intensify heavy precipitation events by increasing the atmosphere's water vapour-holding capacity (Hardwick Jones et al., 2010). According to the CC relationship, water vapour content, and consequently precipitation, increases by approximately 6–7 % K<sup>-1</sup> (Risser and Wehner, 2017). This study investigates to what extent precipitation changes in convective storm events follow the expected CC scaling, and whether deviations from this rate, known as super-CC responses, can be identified under warming conditions.

In our simulations, water vapour content is adjusted according to CC scaling, ensuring an initial increase close to the predicted  $7 \% K^{-1}$  (not shown). However, accumulated precipitation does not consistently follow this trend, as shown in Fig. 9. In particular, Case case 1 and Case case 3 exhibit significantly distinctly greater increases in precipitation, far exceeding  $7 \% K^{-1}$ .

**Figure 8.** (a) Mean hailstone size distribution in the MOSES domain for cases 1 and 2 and the Germany domain for case 3, for all simulation temperatures using the continental CCN concentration (C3) for case 1. The domains are defined in Fig. 1. Line colours indicate the different temperature scenarios. (b, c, d) Dominant hailstone sizes for different temperature regimes and CCN concentrations for (b) case 1, (c) case 2, and (d) case 3.

On 22 June 2023For case 3, precipitation increases by more than 17 % K<sup>-1</sup>, while on 23 June 2021for case 1, an increase of over 9 % K<sup>-1</sup> was observed is simulated. These extreme increases persist even when only looking at rainfall and not total precipitation, suggesting the presence of super-CC scaling effects. In contrast, on 28 June 2021in case 2, precipitation follows the expected CC scaling, increasing by 6.5 % K<sup>-1</sup>. One possible explanation for the strong deviation on 22 June 2023 in case 3 is the presence of a pronounced synoptic front moderate to strong large-scale synoptic ascent supporting convection initiation (see Fig. 2e). In line with previous findings, such as those by Fowler et al. (2021), stronger synoptic-scale ascent and enhanced moisture convergence under warmer conditions may amplify convection and contribute to super-CC responses. This indicates that while CC scaling explains parts part of the observed precipitation increase, additional dynamical and microphysical processes must be considered to account for also contribute to the deviations beyond the thermodynamic expectation for Cases

cases 1 and 3. Further investigation is required 3, necessitating further investigation to identify the mechanisms driving these super-CC precipitation trends. The role of such processes is elaborated in the following sections, non-linear character of microphysical processes and the chaotic nature of atmospheric convection, in which small differences in environmental conditions can lead to large impacts on simulated storms, are likely key factors behind the super-CC scaling. Convection triggering is also highly dependent on environmental conditions; small differences in temperature or humidity can determine whether or not deep convection is initiated at a given time and location. For example, different CCN concentrations can determine whether or not a supercell is successfully simulated (Barthlott et al., 2024). The strong invigoration of convection with temperature has been discussed to amplify the scaling rate with temperature or shifts from low-intensity stratiform rainfall to higher-intensity convective rainfall (?). It should be noted that the short predictability timescale of convective processes, particularly during initiation, can lead to non-linear responses. This inherent sensitivity may contribute to some of the amplified or variable trends observed in the simulations.

Although all PGW approaches maintain relative humidity at model initialisation and specific humidity increases according to the CC relationship (e.g. Kröner et al., 2017), regional climate simulations of Feldmann et al. (2025) have also shown a decrease in frequency of supercell thunderstorms over the Iberian Peninsula and southwestern France which are subject to strong surface warming and relative humidity decreases in the course of the simulations. Such long-term effects cannot be included in our study because our interest is in a detailed process understanding of aerosol—cloud interactions with high-resolution simulations of individual cases with a sophisticated microphysics scheme and explicit hail particle class.

## 3.3 Convective Cell Frequency and Lifespan




To further analyse the occurrence of convective cells, version 1.5.3 of the Python package Tobae (Tracking and Object-Based Analysis of Clouds) (Heikenfeld et al., 2019; Sokolowsky et al., 2024) is used. Tobae is a software to identify, track, and analyse clouds and other meteorological phenomena such as updraughts, precipitation, and radiation. The tool uses algorithms to identify features and link them into consistent trajectories. In this study, updraughts of cells at high temporal resolution () between and are tracked. For the updraught tracking, first, the data of the maximum updraught between 3 and height and the total condensate mixing ratio, which means the total amount of liquid and frozen water per mass of dry air, is calculated. Afterwards, several thresholds () are used to track updraughts to better resolve overlapping cells. In addition, only updraughts with a minimum area of 10 grid cells are used as features, and a maximum distance of 25 grid cells is selected when linking features. As a segmentation threshold, a value of is chosen as the condensate mixing ratio to identify the cloud volumes corresponding to the individual identified updraughts (Heikenfeld et al., 2019). To track only longer-lived cells, only tracks with a minimum lifetime of are used for the evaluation. The simulated cells are further analysed by investigating—

To further analyse the occurrence of convective cells, the Tobac tracking tool was used as described in section 2.4.3 to assess how their number and lifetime are affected by temperature changes in the atmosphere (Fig. 10). For brevity, maps of cell tracks from the numerical sensitivity experiments are omitted, and the analysis focuses instead on the average number of cells and their lifetimes. All days show an increase in the mean number of convective cells and a decrease in their mean lifetime in simulations with higher temperatures. The increase in the number of cells is quite substantial and ranges between

**Figure 9.** Percentage increase in total daily precipitation over the Germany domain and corresponding best-fit lines, compared to the expected Clausius–Clapeyron scaling of  $7 \% K^{-1}$ . Solid lines represent the percentage increase in precipitation over the entire day, while the dashed lines in matching colours show the respective best-fit trends. The domain is defined in Fig. 1.

+37 and +63 % for a +4 K warming scenario, whereas the mean lifetimes are reduced by 5–20 mins only. These averages were computed using all detected convective cells, not only supercells. After investigating precipitation rates, an explanation for the shorter lifetimes of convective cells is found. Precipitation is found to start simultaneously across all investigated temperatures, with higher temperatures leading to greater peak intensities, followed by an earlier decline. This behaviour can be explained by the fact that higher temperatures enhance convective intensity, leading to a faster depletion of the available CAPE, even when overall CAPE is higher. This interpretation is supported by the observed increase in up- and downdraught velocities (Fig. 4), ultimately resulting in shorter cell lifetimes.

## 3.4 Aerosol-cloud interactions (ACI)


To gain an overview of how precipitation is affected by CCN concentration, changes in surface precipitation and total column-integrated parameters such as cloud water, rain, ice, snow, graupel, and hail are analysed (Fig. 11). Additionally, process rates of microphysical mechanisms are examined to explain the changes that have been observed. This analysis focuses on the results from the reference temperature simulations, as all temperature variations exhibited similar behaviour and, therefore, do not require a separate discussion. Following this analysis, the effects of temperature and CCN concentration on cold and warm rain formation processes, as well as on precipitation efficiency, are discussed.

**Figure 10.** Number of tracks detected by the Tobac tool for (a) case 1, (b) case 2, and (c) case 3, and mean lifetimes of convective cells for all temperature regimes and CCN concentrations for (d) case 1, (e) case 2, and (f) case 3. All values are calculated between 06:00 UTC and 24:00 UTC. The MOSES domain was used for cases 1 and 2, while the Germany domain was used for case 3. Line colours indicate the different CCN concentrations. The domains are defined in Fig. 1.

# 3.4.1 Microphysical processes


Generally, a higher CCN concentration results in suppressed precipitation formation due to the more numerous but smaller cloud droplets, making the collision and coalescence process less efficient (Fan et al., 2017). Despite this initial suppression, the cold-phase convective invigoration theory suggests that delayed precipitation at high CCN levels can ultimately lead to more intense convection and stronger rainfall, along with the formation of more cold-phase hydrometeors (Rosenfeld et al., 2008). In contrast, lower CCN concentrations typically lead to fewer but larger cloud droplets, facilitating more efficient precipitation processes (Yau and Rogers, 1996).

In all three simulated cases, total precipitation decreases for higher CCN concentrations. Rain does not exhibit a clear trend, graupel increases, and hail shows a significant large decrease of more than 200% under higher CCN conditions (Fig. 11a-d). Higher CCN concentrations are expected to increase the number of cloud droplets while reducing their size, which may not always lead to a higher total cloud water mass. However, enhanced droplet nucleation can still result in increased columnintegrated cloud water due to suppressed precipitation processes, a pattern that is observed across all three cases (Fig. 11f). Additionally, a decrease in the autoconversion and accretion processes is observed (Fig. 11e, g), both contributing to rain formation, explaining the decrease in rain within the atmosphere. At the same time, evaporation is enhanced for low CCN concentrations (Fig. 11i), which could explain why the increase in column-integrated rain for low CCN concentrations does not result in an increase in surface rain on all three days. Moreover, total column-integrated ice and snow (Fig. 11j, I) increases under polluted conditions, aligning with the cold-phase convection invigoration theory. This theory suggests that delayed pre-

**Figure 11.** Percentage deviations of spatiotemporal averages of autoconversion (AC), accretion (ACC), evaporation (EVAP), rain freezing (RF), melting (MELT), total riming (RIM), deposition (DEP) (left column) and of total column integrated cloud water (tqc), rain (tqr), ice (tqi), snow (tqs), graupel (tqg) and hail (tqh) amounts (right column) from the respective reference run C3 with continental CCN assumption. All points refer to the reference temperature scenario without increments. The Germany domain was used, which is defined in Fig. 1.

**Figure 12.** Vertical profiles of spatio-temporal averages of hail content in the MOSES domain for different CCN concentrations on 23 June 2021 (case 1). The domain is defined in Fig. 1.

cipitation formation allows cloud water to ascend to higher altitudes, transitioning into ice and other frozen hydrometeors. However, the cold-phase convection theory also predicts enhanced convection, which is not supported by the observed updraught intensities (not shown). Instead, a reversed trend emerges, with stronger up-and-downdraughts occurring at lower CCN concentrations.




These findings eall into question raise questions about the applicability of the cold-phase convective invigoration hypothesis in this context. While the theory proposes that increased aerosol concentrations enhance storm intensity through latent heat release during freezing, its overall validity remains uncertain. Numerous studies report contradictory results depending on cloud type, environmental conditions, and the choice of microphysical schemes (van den Heever et al., 2011; Altaratz et al., 2014b; Fan et al., 2017). Recent theoretical work even suggests that aerosol-induced updraught enhancements may be significantly much weaker or even negative than previously assumed (Altaratz et al., 2014b; Igel and van den Heever, 2021). Rain freezing, deposition, and riming are all processes contributing to graupel and hail formation (Fig. 11k, q, o). The only hail production component that decreases significantly strongly decreases with higher CCN concentrations, mirroring the trend in surface hail content, is total riming, which exhibits an average change between 20 and 40 %. However, riming is the most significant important contributor to hail and graupel formation, with a magnitude approximately four times greater than rain freezing and deposition. Consequently, rain freezing and deposition have a negligible impact on hail formation. Interestingly, while column-integrated graupel content decreases in a high CCN environment within the column (Fig. 11n), hail content increases (Fig. 11p), which is the opposite of what occurs at the surface. Typically, a decrease in the total column would not be expected to correspond with an increase at the surface. To understand this behaviour, it is essential to consider the differences in hailstone sizes at varying CCN concentrations. This study previously found that hailstones grow larger in low CCN

**Figure 13.** Cold-to-warm rain ratio (DEP+RIM)/(AC+ACC) for different CCN concentrations and temperatures for (a) case 1, (b) case 2, and (c) case 3. For cases 1 and 2, the MOSES domain was analysed for case 3, the Germany domain. The domains are defined in Fig. 1.

environments. Due to Conversely, in low CCN environments, where hailstones are larger, melting is expected to be less efficient due to their lower surface-to-mass ratio, larger hailstones undergo less melting compared to smaller ones. This effect is evident in the vertical profile of atmospheric. This interpretation is consistent with the vertical profiles of hail content (Fig. 12), which shows that in high CCN environments, where hailstones are smaller, increased melting occurs. Conversely, in low CCN environments, where hailstones are larger, melting is significantly reduced. As a result, larger hailstones are more likely to reach the surfaceunder low CCN conditions, suggest reduced melting in low CCN cases (lowest maximum around 4km agl of all different CCN concentrations, but largest values at the ground). It should be noted that the model output provides combined melting rates of hail and graupel, so hail-only melting cannot be quantified separately. Nevertheless, the earlier analysis of hail size distributions at the surface (Fig. 8) clearly shows that larger hailstones reach the ground under low CCN concentrations, and the reduced efficiency of melting provides a plausible mechanism for this behaviour. A comparable mechanism operates in warmer environments: stronger updraughts favour the growth of larger hailstones, which are less sensitive to melting due to their lower surface-to-mass ratio, while smaller stones are more prone to melting during descent. This dual effect explains why both warming and CCN reductions contribute to a coherent shift in the hail size distribution towards larger hailstones at the surface. Simultaneously, more graupel transforms into hail, while the remaining graupel melts before reaching the surface, leading to a decrease in surface graupel content under lower CCN conditions. This mechanism could also explain the overall relatively constant melting rates -(Fig. 11m). However, since the output of the model does not differentiate between the melting rates of graupel and hail, direct verification of this hypothesis remains challenging.

### 3.4.2 Cold Rain formation processes and warm rain precipitation efficiency





Rain can form through warm and cold processes. Given that the simulations are done for various temperatures and CCN concentrations, it is anticipated that the ratio between cold and warm rain formation will differ. This ratio is determined by dividing the contributions of processes that generate cold and warm rain, respectively, as follows:

$$\frac{\text{cold rain}}{\text{warm rain}} = \frac{\text{DEP} + \text{RIM}}{\text{AC} + \text{ACC}}$$
 (6)

DEP represents deposition, RIM denotes riming, and AC and ACC are autoconversion and accretion (Barthlott et al., 2022a). Figure 13 presents the results for each simulation day, showing variations in temperature and CCN concentrations. Across all days, higher CCN concentrations are associated with an increased cold-to-warm rain formation ratio, indicating a stronger contribution of cold rain processes. In contrast, higher temperatures lead to a reduction in this ratio. Note that the vertical axis for 22 June 2023 differs from the others, possibly due to the presence of strong synoptic forcing on that day. From a physical perspective, high concentrations of aerosols slow down the onset of warm rain by inhibiting autoconversion and lowering the effectiveness of collision—coalescence. Consequently, a larger amount of cloud water can ascend into the mixed-phase region, where it contributes to cold-rain formation. Consequently, a shift to the cold pathway exists. In the scenarios with higher temperatures, the cold-to-warm ratio is systematically reduced across all CCN regimes. The higher the temperature increment, the lower the resulting cold-to-warm ratio. As expected with higher freezing levels in warmer environments, the vertical range for the warm-phase pathway increases, whereas the one for the cold-rain pathway decreases. As a result, the percentage increase with larger aerosol loads is smaller for higher temperatures.

Figure 14a displays the ratio of cold-to-warm rain formation processes, as a function of precipitation rate for the 23 June 2021 case . case 1. The ratio increases systematically with CCN concentration, from approximately 1.9 for C1 to over 4.1 for C4. This represents more than a twofold increase in the relative contribution of cold rain processes when moving from the lowest to the highest CCN scenario. This trend highlights the strong underscores the pronounced sensitivity of rain formation mechanisms to CCN concentrations: higher CCN concentrations suppress warm rain formation through delayed autoconversion and enhanced cloud droplet competition, thereby shifting precipitation generation toward cold processes, such as deposition and riming. Importantly, the ratio remains remarkably stable across a broad range of precipitation rates above 0.2 mm per 30 min. This stability above a threshold suggests a robust suggests a shift in microphysical regimes primarily driven by CCN concentration, consistent with findings from Barthlott and Hoose (2018).

#### 3.4.3 Precipitation efficiency

Another parameter important Another important parameter for precipitation formation is the precipitation efficiency, which precipitation efficiency (PE). Total surface precipitation (P) is determined by both the generation of hydrometeors (G) and the fraction of this condensate that ultimately reaches the ground, described as PE. PE is defined as the ratio of the precipitation P and precipitation to the generation term P, including,

$$PE = \frac{P}{G} \tag{7}$$

where G includes all processes that generate condensates in the atmosphere(Baur et al., 2022):

$$PE = \frac{P}{G}$$






The generation processes considered in this analysis include the microphysical processes of, namely deposition, riming, autoconversion, and accretion—(Baur et al., 2022). Analysing PE provides additional diagnostic insight into how efficiently clouds convert condensate into surface precipitation. In particular, PE highlights the role of microphysical processes such as

**Figure 14.** (a) Cold-to-warm rain ratios (DEP+RIM)/(AC+ACC) plotted for each half an hour for different CCN concentrations (case 1): C1 (blue cross), C2 (orange circle), C3 (green triangle), C4 (red hexagon). Dashed lines in the corresponding colour represent the mean taken for all data points having precipitation rates larger than 0.2 mm / 30 min. The legend includes the mean cold-to-warm rain ratios of each CCN concentration shown by the dashed lines. (b) Precipitation efficiency of all four CCN concentrations for all simulation temperatures for case 1. The MOSES domain was used, which is defined in Fig. 1.

evaporation and melting, which cannot be inferred directly from P or G alone. Fig. ?? 14b shows the calculated precipitation efficiency (PE) on 23 June 2021-PE for case 1 and the sensitivity to varying CCN concentrations, reflecting the relationship between condensate production and the amount of precipitation reaching the surface. An increase in PE is observed with higher CCN concentrations, indicating that a larger fraction of the formed hydrometeors reaches the surface. The generation term G (not shown) was also analysed separately and shows a decreasing trend with increasing CCN concentrations. Since total precipitation P depends on both PE and G, the net effect varies: if the increase in PE outweighs the decrease in G, precipitation increases, and vice versa. In most simulations, precipitation decreases with higher CCN levels (not shown), except for Case case 3, where precipitation remains nearly constant at +2 K and even increases at +3 K. Overall, the changes in precipitation remain relatively modest, typically within 10 %.

## 4 Summary and Conclusions




This study investigated how convective storms in Central EuropeGermany, especially supercells, would evolve under future climate conditions and environments with different aerosol loads. To explore these influences, this study conducted convection-resolving convection-permitting real-case simulations of supercell events previously observed during the Swabian MOSES field campaigns of 2021 and 2023. These events were simulated across five temperature scenarios, including a reference simulation, +1, +2, +3, and +4 K, and four different CCN concentrations to observe storm behaviour under varied environmental conditions. The different CCN concentrations included values typically observed in Germany, providing a reliable reference point for comparison (Hande et al., 2015). The The temperature

perturbations were applied using the PGW approach, which isolates thermodynamic climate signals by consistently modifying initial and boundary conditions. This methodology enables a controlled investigation of the sensitivity of convective storm characteristics, such as storm intensity, duration, and convective precipitation, to future warming. By employing high-resolution, convection-resolving simulations on a 1 km grid, this research enables a detailed examination of convective processes, a depth of analysis that cannot be found in similar studies. This high-resolution model enables a comprehensive understanding of the dynamics of supercells and storm intensities in response to changing environmental conditions.

The results demonstrate that higher temperatures generally increase convective parameters such as CAPE and CIN, leading to more intense convection and a greater number of convective cells overall. This, in turn, accelerates the depletion of CAPE. The shorter lifetimes of convective cells in warmer environments further illustrate this effect. Other papers, such as Huang et al. (2024); Mallinson et al. (2024); Yang et al. (2024), also observed an increase in CAPE and CIN, suggesting a more favourable environment for stronger convective systemsand less favourable ones for weaker systems. This combination implies that stronger triggering mechanisms are required to initiate convection, which in turn would favour the development of fewer but more intense convective systems.





Another important finding was the correlation of larger mean helicity values with increased atmospheric temperatures. Helicity values serve as indicators for updraught rotation and, therefore, supercells. Therefore, the simulations demonstrated that in a warmer environment, the likelihood of supercell development is increased, which leads either to more supercell formation, more intense supercells, or a combination of both. This result gained by single case studies agrees well with recent climate simulations of Feldmann et al. (2025). Comparing a current-climate simulation with a pseudo–global warming scenario (+3 K), the future climate simulation shows an average increase of supercell occurrence by 11 %. Furthermore, updraught intensity is increased in warmer environments, which supports the hypothesis of stronger convective cells.

The study also identified significant large changes in precipitation amounts and the spatial distribution of rainfall in response to varying temperatures. As temperatures increased, total precipitation amounts generally rose, with both rain and hail exhibiting greater amounts. The analysis of the 95th percentile and the area that is affected by the 95th percentile both indicated a rise, which heightens the risk of flash floods. Additionally, for two of the three simulated cases, a super-CC scaling rate was observed, which means that the increase in precipitation can only be explained by additional factors beyond the increase in water vapour in the air column. Further analysis of hail characteristics indicates that higher temperatures and lower CCN concentrations, both anticipated in a future climate, lead to an increase in hailstone size, particularly for mean and large hailstones during convective events. This trend is especially concerning given the significant high damage potential of larger hailstones. The results further indicate that warmer environments foster stronger updraughts, which support the growth of larger hailstones. These larger stones are less affected by melting due to their lower surface-to-mass ratio, while smaller ones are more efficiently lost during descent. This dual process is consistent with previous findings (Raupach et al., 2021) and explains the coherent shift in hail size distributions toward fewer small and more large hailstones under climate change.

The analysis of how ACI is influenced by varying CCN concentrations influence ACI revealed several notable effects. Importantly, the influence of CCN on microphysical processes remains consistent across all temperature regimes. In high CCN environments, the processes of autoconversion and accretion, which are critical for warm rain formation, decrease, explaining

the observed decrease in surface rainfall. The previously discussed decrease in hail was then explained by-attributed to a reduced riming process, which is the dominant-primary mechanism for hail formation. A closer examination of hail dynamics showed that there is more hail within the atmospheric column in environments with higher CCN concentrations. However, the amount of hail decreases at the surface compared to simulations with lower CCN concentrations. This counterintuitive result was explained by the smaller size of hailstones formed in high CCN conditions, which increases melting rates compared to the larger hailstones produced in low CCN environments. These The differences in formation processes result in an increased cold-to-warm rain formation ratio in environments with higher CCN concentrations and a decreased ratio in higher temperatures. Moreover, precipitation efficiency was considerably increased in environments with high CCN concentrations. As a result, while fewer hydrometeors were produced in high CCN concentration settings, a larger fraction of them reached the Earth's surface. These findings indicate




Due to the small number of case studies analysed here, no general conclusions can be drawn. Furthermore, our PGW approach, with uniform vertical warming and maintained relative humidity, does not account for possible lapse rate changes (e.g. Brogli et al., 2021), shifts in weather patterns at the synoptic level, or humidity gradients that may develop over longer simulation periods. However, our results, obtained using a sophisticated double-moment microphysics scheme that includes explicit hail, provide valuable insights into the effects of aerosols on mixed-phase clouds in various warming scenarios. The findings suggest that convective storms, especially supercell events, will likely particularly supercells, are likely to intensify, resulting in increased rainfall heavier precipitation and larger hailstones. This intensification The increasing intensity of precipitation hazards raises the risk of flash floods and poses a greater threat to property, agriculture, and infrastructuredue to the heightened potential for severe weather impacts. While this study provides valuable insights into the response of supercell storms to warming and aerosol changes, it is based on a limited set of three case studies. Additionally, the simulations assume constant relative humidity and apply only homogeneous warming, without considering vertical variations in temperature that would affect atmospheric stability. Large-scale dynamical changes are also not included. Despite these constraints, the results reveal clear and consistent trends, highlighting the robustness of the findings.

*Code availability.* This study is based on the ICON model version 2.6.6. The ICON model is available to the community under a permissive open source licence (BSD-3C) at https://gitlab.dkrz.de/icon/icon-model (DWD et al., 2025).

Data availability. The nature of the 4-D data generated in running the model experiments requires a large tape storage facility. These data are of the order of 200 TB (terabytes). The model data can be made available by the authors upon request. Global Forecast System (GFS) data is available through https://www.ncei.noaa.gov/products/weather-climate-models/global-forecast. The data needed to replicate the figures can be found at https://doi.org/10.5281/zenodo.15830656 (Lucas et al., 2025).

Author contributions. CB developed the project idea and designed the numerical experiments. LL performed the numerical simulations and conducted the analyses. LL wrote the paper with contributions from all co-authors.

Competing interests. The contact author has declared that none of the authors has any competing interests.



Acknowledgements. The authors thank the German Weather Service (DWD) for providing the ICON model code, the initial and boundary data, and the RADOLAN data. The field campaigns Swabian MOSES 2021 and 2023 were supported by funding from the Helmholtz Association within the framework of MOSES. The authors gratefully acknowledge the computing time made available to them on the high-performance computer HoreKa at the NHR Centre NHR@KIT. This centre is jointly supported by the Federal Ministry of Education and Research and the state governments participating in the NHR (www.nhr-verein.de/unsere-partner). This work was performed with the help of the Large Scale Data Facility at the Karlsruhe Institute of Technology, funded by the Ministry of Science, Research and the Arts Baden-Württemberg and by the Federal Ministry of Education and Research. The editing process was supported by OpenAI's GPT-4, which was used to suggest revisions for language, structure, and readability. "Grammarly" generative AI (https://app.grammarly.com/) was used for the whole article's grammar, spelling, and stylistic check.

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
