# Peer review of "Aerosol effects on convective storms under pseudo-global warming conditions: insights from case studies in Germany"

_EGUsphere, 2025_

## Referee Comment (RC1)

**Review for "Pseudo-Global Warming Simulations Reveal Enhanced Supercell Intensity and Hail Growth in a Future Central European Climate" by Lucas et al., ACPD**

**Summary**

This study investigates the response of convective storms to CCN perturbations and a warming climate, with the help of a so-called pseudo-global warming approach. The authors selected three case studies of convective storms passing over Germany and employed the ICON model in 1 km horizontal resolution with a two-moment microphysics scheme. A strong focus was put on changes in supercells, hail growth, and the underlying microphysical processes within these clouds. Overall, global warming has a dominant effect on the evolution of convective storms, with aerosol–cloud interactions only playing a secondary role. The authors highlighted several aspects important to convective storms and their evolution. However, by having this holistic approach, I believe some structured and in-depth discussions were lost. I also think the authors should try to combine more of their findings in a concise explanation and figures, which currently seem more like a checked-off list. I am detailing my major and minor comments below.

**Major comments**

- **Case studies:** I have several questions regarding the selection of the case studies. First of all, the selected days are motivated by the MOSES campaign in Germany, and often the analysed domain is focused on that part of Germany. However, no comparison to observations were done, so why this underlying motivation? If the MOSES campaign is important, then why was the third case selected? It falls out of line in several aspects throughout the analysis and it makes it difficult to fully grasp the important aspects of the findings. This is, e.g., illustrated by varying y-axes in Figure 6, 10, and 13. Is the third case really helping the analysis? Because the authors often attribute the deviations to the synoptic forcing, which to some extent is not satisfying. Moreover, it is often not clear to me what part of the domain was now analyzed for which part, which ideally should not change (i.e., Germany or MOSES) because it also makes the comparison between the results trickier. If the MOSES domain is not crucial, I would recommend that for all results the whole domain of Germany is analyzed. If the MOSES domain is crucial, then the third case study does not contribute much to the scientific findings. Also, often figures are only for one case, and the next figure shows again all cases. A more consistent approach here would be helpful. Either first doing an overview for all three and then selecting one, or show all three cases in detail.

  Regarding the clarity of the writing: The authors switch between the naming of the cases, i.e., Case 1-3 or the dates, and the date format changes. I would highly recommend to find a clear naming convection and stick to that. Also Figure 2 could be improved if the border of Germany and the MOSES domain are highlighted in the left columns of subfigures.

- **Significance:** Given the set of ensembles for each case study, I am missing a quantification of the uncertainty of all the investigated quantities / processes. In the text, the authors often say these changes are significant or robust without offering much evidence. While there may be systematic changes, they do not necessarily have to be significant. I would recommend to conduct a proper statistical analysis for each investigated variable. This is notable, e.g., in Figure 5, but also others, often showing only the mean over a differing areas (Germany vs. MOSES). Also in Line 354 the authors state that the hail production significantly increases with CCN, but how was this assessed? With that I disagree with the concluding remark, that these findings are robust (which they very well may be) without a proper assessment.

  Another question I had is the impact of the windows for the moving averages for the cold pools. Given that the storms do not live longer than 10 hours (according to Figure 3), how is a temporal window of 8 justifiable, and also where do the 166 grid points come from?

- **Storyline:** The authors often talk about results and indicate "not shown". I do understand that not all results can be shown, but here I expect that a concrete storyline is built, which may also combine different figures into one, to not overwhelm the reader. However, the result sections actually starts with results that are not shown, which is not satisfying at all. Regarding the CAPE and CIN values, the authors could think of a table summarizing these numbers, or actually using the Appendix, but I would argue, that results that are not shown are definitely not the first result to be discussed. Furthermore, I believe the hail sections and their figures can be combined into one, as especially for hail the size is more important than the mass mixing ratio. Here, I actually do not understand Figure 7, as ICON with two-moment microphysics can well simulate larger hailstones than 1 cm in diameter.

What limits are discussed here and what curves have been extended? I do not see that in Figure 7. Restructuring the manuscript and the figures requires time, but this could be helpful to clearer convey the scientific findings.

Regarding the introduction: here the sections on PGW (what exactly is perturbed in the IC and BCs mentioned in Line 29?) and convective invigoration should be extended, as in both important aspects and references are only mentioned later in the manuscript. Especially the convective invigoration is a highly debated topic, as the authors correctly state later in the manuscript, but they do not provide the proper context already in the introduction. Within the convective invigoration discussion, I would have loved to actually see changes in number concentrations of the hydrometeors, which to my surprise is not discussed at all, even though a two-moment microphysics scheme is employed. Moreover, the latent release heat could be quantified at least in terms of temperature changes within the cloud if the respective diagnostic is not available in the already produced model output.

Regarding the conclusions: To me, they read more like a summary than a conclusion, as many aspects are repeated and not presented in a concise manner with clear take-home messages.

**Minor comments**

- Line 4: the acronym ICON is missing.

- Line 5: aerosol effects on what? from context it is clouds, but it should be specified

- Line 20: "the three-day period around 23 June 2021" could be explicitly stated: Is it from 22-24? Or 21-23?

- Line 23: A reference would be nice for this event on 23 June 2021

- Line 86: it is not a triangular grid, but a icosahedral grid, which is trisected forming the triangles

- Line 87: What is the height of the model top? This is an important information to have a better understanding of the vertical resolution.

- Line 88: SLEVE coordinates were introduced by Schär et al. 2002, so the original source should be cited

- Line 108: Surface temperatures are adapted accordingly in what way? Is there a formula behind it, or is the, e.g., +1 K imposed? A clearer explanation would be great.

- Line 112: I believe that 1700 cm$^{-3}$ CCN are rather on the high side, as also shown by Schmale et al. 2018. It is fine to use the hardcoded values in the two-moment microphysics scheme, but I believe the chosen concentrations should be better contextualized. As far as I know, these concentrations emerge from Segal and Khain, 2006, and are based on some rather early measurements of CCN.

- Line 150: wrong chapter reference → should be 3.2?

- Line 174: "To identify ..." sentence is doubled with the next sentence of the new paragraph.

- Figure 6 caption: what is meant by mm for the rain and hail? Is it a precipitation rate?

- Figure 8: Adding the numbers to the single tiles of the heatmap would help to grasp the figure in a faster way.

- Line 292-294: These two sentence basically say the same things, and can be combined.

- Figure 10 caption: the domainS ARE defined ...

- Figure 11 caption: The ending in the caption, saying the reference simulations are with C3 does not make sense to me here. All CCN variations are shown in the plots for all integrated quantities, or am I misunderstanding something?

- Figure 12: Where are these vertical profiles coming from? Are these based on mean values? The hail content seems very low.

- Line 342: what is the 1 in (Fig. 11j, 1)?

- Line 364: I am wondering if the conclusion of "larger hailstones are more likely to reach the surface under low CCN concentrations" is circular, because under low CCN, the hailstones are larger. What is more important? So, the question is how does the melting rate change with CCN? From Figure 11, it looks more like that the melting rate is rather independent of the CCN concentration. I do not think that the sentence here is then fully correct.

- Line 368: A reference to the figure 11 should be made again from the melting rates.

- Line 385: The ratio of warm and cold rain and its dependence to the precipitation intensity is discussed in the next section, but the authors already mention it here, without giving much context or justification. I would move that to later.

- Line 404: I disagree with the statement that this study investigates convective storms in Central Europe, because it does it only for Germany. This also then connects to the title, which in my opinion is misleading, as in principal Germany was looked at. The authors should consider adapting their title to that.

- Line 408: I also disagree with the statement that the chosen CCN concentrations are typical for Germany, which is not the case. These concentrations are coming from Segal and Khain, as elaborated above.

- Line 416: The results do not demonstrate any increase in CAPE and CIN as the authors decided to not show this. I elaborated my reasons for not doing this above.

- Line 448: The sentence "This intensification ..." is circular. Removing the last part "due to the heightened ..." would remedy that.

**Editorial comments**

- Line 10: wrong / incomplete latex command for the unit

- Line 173: plotting is colloquial and should be avoided

- Line 208: it is unusual to me to denote mixing ratios with $r$, maybe the authors can consider opting for a more common writing style

**References**

Schär, Christoph, Daniel Leuenberger, Oliver Fuhrer, Daniel Lüthi, and Claude Girard (2002). "A New Terrain-Following Vertical Coordinate Formulation for Atmospheric Prediction Models". In: *Monthly Weather Review* 130.10, pp. 2459–2480. ISSN: 1520-0493, 0027-0644. DOI: 10.1175/1520-0493(2002)130<2459:ANTFVC>2.0.CO;2.

Schmale, Julia, Silvia Henning, Stefano Decesari, Bas Henzing, Helmi Keskinen, Karine Sellegri, Jurgita Ovadnevaite, Mira L. Pöhlker, Joel Brito, Aikaterini Bougiatioti, Adam Kristensson, Nikos Kalivitis, Iasonas Stavroulas, Samara Carbone, Anne Jefferson, Minsu Park, Patrick Schlag, Yoko Iwamoto, Pasi Aalto, Mikko Äijälä, Nicolas Bukowiecki, Mikael Ehn, Göran Frank, Roman Fröhlich, Arnoud Frumau, Erik Herrmann, Hartmut Herrmann, Rupert Holzinger, Gerard Kos, Markku Kulmala, Nikolaos Mihalopoulos, Athanasios Nenes, Colin O'Dowd, Tuukka Petäjä, David Picard, Christopher Pöhlker, Ulrich Pöschl, Laurent Poulain, André Stephan Henry Prévôt, Erik Swietlicki, Meinrat O. Andreae, Paulo Artaxo, Alfred Wiedensohler, John Ogren, Atsushi Matsuki, Seong Soo Yum, Frank Stratmann, Urs Baltensperger, and Martin Gysel (2018). "Long-Term Cloud Condensation Nuclei Number Concentration, Particle Number Size Distribution and Chemical Composition Measurements at Regionally Representative Observatories". In: *Atmospheric Chemistry and Physics* 18.4, pp. 2853–2881. ISSN: 1680-7324. DOI: 10.5194/acp-18-2853-2018.

---

## Referee Comment (RC2)

**Review - Pseudo-Global Warming Simulations Reveal Enhanced Supercell Intensity and Hail Growth in a Future Central European Climate**

**Summary**

This manuscript presents an number of simulation experiments discussing supercell and hailstorm intensity under different initial conditions, by changing the surface temperature and CCN content. Three case studies are selected and experiments are performed at 4 different warming levels and 4 different CCN concentrations. The authors conclude that supercells become more intense and produce larger hail in warmer conditions. Possible reductions in CCN content due to reductions in pollution further modify these effects.

While this study targets a subject that is highly relevant, it comes with many limitations and the manuscript currently overstates the conclusions that can be drawn from this. Moreover, the work is not well-situated within the broader literature. Established process changes for severe convection under climate change are not discussed properly in the context of the presented results (e.g. how increasing temperature at constant RH implies increased CAPE and CIN).

While the study addresses an important topic and includes interesting case studies from a recent field campaign, I recommend major revisions before it can be considered for publication. The paper has clear potential, but currently suffers from several methodological, interpretational, and structural limitations that should be addressed to strengthen its scientific credibility and clarity. The major points are listed below, as well as line-by-line comments.

These revisions are essential to ensure the study provides a robust and realistic contribution to our understanding of supercell and hail processes and their sensitivity to warming.

**Major comments:**

1. Simulation setup: While the uniform vertical warming with constant RH meets the technical definition of a PGW setup, this is a highly idealized and simplified approach. Combined with the fact, that this manuscript targets 3 case studies, no general conclusions on convective severity in a future climate should be made. Therefore, the title appears overstated with respect to the results provided.
   Instability changes in Europe are complex and tied to far more than uniform vertical warming. Changes in lapse rates and relative humidity are key to appropriately reflecting future instability situations in Europe. As such, the experiments performed here do not necessarily reflect a future climate state, but a generically "increased CAPE and CIN" scenario.
   These limitations are listed almost as an afterthought, in the last paragraph of the manuscript, which is not an appropriate location to detail essential methodological constraints.

2. Validation of simulations: Despite picking case studies from a field campaign, no observational comparisons are conducted. Germany provides both a radar-based hail size estimate, as well as a crowd-sourced report database. Case studies from neighbouring Switzerland of the same day are not referenced, or discussed as observational reference. Precipitation data is also not used as a comparison.

3. Discussion of hail size: The presented mean hail size distributions raise questions as, commonly, hail is considered to have a minimum size of 5 mm, which is the value where most distributions peak here. While Fig. 7 mostly serves to illustrate changes in distributions, commonly hail size distributions are more analyzed with respect to their larger hailstones, not just the mean. And even for the mean, these sizes appear very small. Especially for cases like June 28, 2021, where in Switzerland hail size reports exceeded 8 cm and reports, photos and sensor measurements indicate mean hail sizes well beyond 5 mm. Fig. 8 further shows dominant hailstone size, which still yields sizes well below 1 cm. Without any discussion of the observed hail sizes and a comparison to this, this poses the question, whether a) these cases are representative of damaging hail cases in southern Germany, or b) whether the modeled output sufficiently captures this.

4. Discussion of hail size and climate change: In different parts of the manuscript, the combination of increased updraft strength and melting is discussed inconsistently. It is

established, that higher-CAPE and hotter environments are expected to sort for larger hail sizes. Increased updraft strength and size increases the potential for larger hailstones. These are less affected by melting than the smaller ones. Hence, smaller stones are more likely to melt and decrease in occurrence at the surface, whereas large ones increase. Both of these aspects lead to shifts in the hail size distribution and do not contradict each other. This is not discussed coherently throughout the manuscript and not embedded in the current state of literature.

5. Discussion of super-CC scaling: A brief discussion of where / how the additional precipitation falls would be very desirable in this context. Is it tied to more or longer lived convective storms? Is it tied to trailing stratiform precipitation, perhaps associated with the approaching fronts? Given that cell tracking is already performed on the data, this would provide meaningful additional information and reduce the remaining hypotheses in the manuscript.

6. Discussion of INP vs CCN: The modeling experiments only touch upon the sensitivity towards CCN concentration. However, for hail formation pathways, INP concentration is a key element. E.g., hail seeding experiments aim to target INP concentration, rather than CCN concentration. At least a discussion of the conditions assumed for INPs / freezing processes should be included.

7. Differentiation of supercells from all convection: The study claims that supercells in particular are intensifying and producing more hail. From my understanding, most analyses were conducted in a set model domain and not stratified by storm type. This should either be clarified, or the interpretation should be adapted.
   Given the importance of shear for convective organization and supercell development, the shear conditions should also be discussed somewhere. I am aware that these experiments do not systematically modify the initial wind field, however it is implied several times that the synoptic forcing plays out differently in the warming scenarios, which would, in turn, also affect the vertical shear profile and the potential for convective organization.

8. Clarity of results: While the study follows 3 case study, single cases are highlighted at different points and it is not quite clear, why which case is focused on where. Overall, the manuscript lacks a clear storyline and presentation, seeming more like a list of results. This does not require large content changes, but rather an adaptation of the framing to make the main points more clear to the reader. Personally, I would find it helpful to have all methods in the methods section, as opposed to the beginning paragraph of each results block.

9. Missing literature discussion: A significant portion of recent severe convection literature is not referenced or discussed in this study. A list of relevant studies is provided below, with a short note of their respective context. This contains a certain amount of Swiss literature, as the selected region is very close to the border and these studies generally extend into southern Germany, to the extent that the Swiss radar network provides coverage.

- https://rmets.onlinelibrary.wiley.com/doi/full/10.1002/wea.4306
  Detailed case study of June 28, 2021 in Switzerland (synoptic discussion, radar data and hail reports include southern Germany)
- https://wcd.copernicus.org/articles/6/645/2025/
  Ensemble model study of June 28, 2021
- https://arxiv.org/abs/2503.07466 (in press)
  Modelled current and future climatology of supercells in Europe, including hail size and environmental analysis (see heterogeneous evolution of instability in Europe)
- https://www.nature.com/articles/s41558-023-01852-9
  Wind gust and downdraft change with climate change (thermodynamical reasoning for increasing downbursts, relevant for Fig. 4)
- https://egusphere.copernicus.org/preprints/2024/egusphere-2024-3924/ (accepted)
  Influence of Sahara dust on hailstorms in Europe
- https://journals.ametsoc.org/view/journals/mwre/152/2/MWR-D-22-0350.1.xml
  Topographic effects on supercells, including experiments with varying CAPE
- https://agupubs.onlinelibrary.wiley.com/doi/abs/10.1029/2024JD042828
  Modeled hail climatology of Europe

- https://www.authorea.com/doi/full/10.22541/au.173809555.59545480
  Modeled future hail climatology of Europe, including associated precipitation and environmental changes (see also moisture trends)
- https://www.nature.com/articles/s41612-023-00352-z
  Observational reference of severe thunderstorm types – including how many supercells have hail and vice versa
- https://egusphere.copernicus.org/preprints/2025/egusphere-2025-918/ (in press)
  Changing hailstorm environments with climate change
- https://journals.ametsoc.org/view/journals/apme/62/11/JAMC-D-22-0195.1.xml
  Hail trends based on environmental changes (ERA-5)
- https://www.researchsquare.com/article/rs-6196143/v1
  Very large hail trends
- https://www.sciencedirect.com/science/article/pii/S0169809520311224
  https://www.sciencedirect.com/science/article/pii/S0169809515002719
  Supercell climatology Germany
- https://www.sciencedirect.com/science/article/pii/S0169809516306020
  Hailstorms and supercells in Germany
- https://agupubs.onlinelibrary.wiley.com/doi/full/10.1029/2019RG000665
  Review on relevant processes for hail formation and hail in a changing climate
- https://link.springer.com/article/10.1007/s00382-024-07227-w
  Modeled European hail climatology
- https://amt.copernicus.org/articles/17/7143/2024/amt-17-7143-2024.html
  Normalization of hail size number distributions (possibly a useful reference for the conversion of microphysical output to HSD)
- https://wcd.copernicus.org/articles/2/1093/2021/wcd-2-1093-2021.html
  Expected changes in summer lapse rates

**Line-by-line remarks:**

- All figures: The standard color scheme used for all discrete color classes is not very color blind friendly. I would recommend to switch to something at least without red-green contrast, such as the IBM color scheme (https://lospec.com/palette-list/ibm-color-blind-safe)
- Line 10: format issue with unit[400]%
- Line 63: Acronym ICON not defined / only defined later in line 73
- Line 73: 1 km is not fully convection-resolving, ideally convection-permitting should be used.
- Line 76: acronym 2MOM is only used once, please consider replacing with double-moment
- Section 2.2: While the methodology is laid out in detail here, its limitations need to be discussed more thoroughly. Overall, to embed the manuscript better in the literature, it may be beneficial to include a dedicated discussion section.
- Fig. 1: What was measured in Villingen-Schwenningen and how does it relate to the simulations?
- Fig. 2: The colorbar for precipitation should be cropped around 60
- Line 156: CIN was already defined. Please check this for all acronyms, there are multiple inconsistencies.
- Line 160: CAPE and CIN changes depend heavily on the chosen experiment, especially on changing lapse rates and RH. Maintaining lapse rates and RH basically mandates an increase in both CAPE and CIN, as the moist adiabat steepens, while everything else remains more or less constant. Moreover, the dewpoint increases less than the temperature, leading to an increase in LCL.
- Fig. 3: and corresponding analysis: The value of 75 m2/s2 from Ashley et al., 2023, was established on 4 km resolution data. Updraft helicity is very strongly impacted by horizontal resolution. To have the same criteria, this value needs to be adjusted in relation to the resolution (this is also mentioned in the methods of Ashley et al., 2023). If the same threshold at a higher resolution is desirable, it should be justified differently.
- Fig. 4: I do not fully understand what is depicted in Fig. 4. What exactly is evaluated in the reference and warming simulation? And how are the percentage changes computed?

- Line 199: Why is the CCN sensitivity not shown? At least this should be included in an Appendix or Supplement. The following description of the vertical structure is not very clear without any supporting material.
- Line 204-214: All descriptions of methodology / computations should be contained in the methods section.
- Line 228: Considering that the experiment does not include changes to the large-scale dynamics, could you please elaborate how the synoptic forcing is expected to change?
- Fig. 6: Given the constraint on xmax for the derived hail size distribution, how does this affect the possibility to meaningfully calculate the 95$^{th}$ percentile of the HSD?
- Lines 245-257: Assuming the same method is used to obtain Fig. 6, this methodological explanation should be earlier, ideally in the methods section.
- Line 261: Enhanced melting does not immediately mean shifting the distribution peak towards smaller sizes, as it affects small stones more than large stones and can skew the HSD towards the tail. Moreover, hotter conditions are often related to greater instability and greater updraft speeds, especially in this simulation setup, where RH is kept constant and higher temperatures basically just increase CAPE.
- Line 283: Is a vapor content increase close to CC in line with climate projections for central Europe? While moisture trends have much greater uncertainty than temperature trends, a number of models suggest a drying throughout much of Europe, including southern Germany (e.g. Fig. 4 in Thurnherr et al., 2025). This needs to be discussed somewhere, as a constant RH can not immediately be presumed to be representative of a future climate state.
- Line 289ff: Can you visualize the precipitation associated with the front and with convection to support your hypothesis?
- Section 3.3: The tracking methodology should be detailed in the methods. How are splits and mergers handled, how are they "counted" for the cell number? Tracking algorithms can be quite sensitive to this and hence also the total cell number. The mentioned behavior of faster convective lifecycles may apply to multicell and single-cell thunderstorms, but should not apply to supercell thunderstorms (the initial main focus of this study), as they normally propagate along long tracks, as long as the encountered environment remains favorable. While the listed explanation may apply to less severe storms occurring in the model domain, this differentiation should be made clear.
- Line 361: Here the smaller sensitivity of larger hail to melting is mentioned, but this should be consistently discussed throughout the manuscript sections.
- Line 378: Could the relevance of the synoptic forcing be verified by providing a spatial visualization of the cold-to-warm ratio? The presence of the forcing is mentioned in many hypotheses throughout the manuscript, without attempt to verify them.
- Line 386: Is there a specific reason to choose the unit mm / 30 min?
- Section 3.4.3: It is a bit unclear to me, which point exactly is being made here. If precipitation efficiency increases, but generation decreases and hence total precipitation decreases, why are not all 3 terms discussed in combination here? Why does precipitation efficiency require a separate analysis?
- Line 405: Consider replacing convection-resolving with km-scale or convection-permitting
- Line 416: Increasing temperatures vertically homogeneously, while maintaining constant RH, basically mandates an increase in CIN and CAPE. This is inherent to the simulation setup and not a finding. The CAPE and CIN conditions, as well as their changes are not shown in detail in the first place, results that are not shown should not constitute a main point of the conclusion.
- Line 418ff: Changes like this are highly regionally dependent and should not be generalized to this extent. What is exactly is meant by stronger storms? Which variable?
- Line 421ff: Supercell frequency and intensity are two different matters and should be discussed separately. Frequency overall cannot be addressed by case studies, as the frequency of supercell-favorable days cannot be deduced from this. The simulations have the data necessary to identify, whether UH is increasing in intensity per storm, increasing its area per storm, or if there are more storms. Ideally this should be addressed quantitatively and not left to speculation.

- Line 447: This should be rephrased to reflect the limitations of the PGW setup and the nature of case studies.

---

## Author Comment (AC1)

**Responses to the reviewers**

Pseudo-Global Warming Simulations Reveal Enhanced Supercell Intensity and Hail Growth in a Future Central European Climate

by L. Lucas et al. October 10, 2025

We thank both reviewers for reading the manuscript and providing detailed comments. We have carefully considered all comments and changed the manuscript accordingly. Please find below our responses in blue.

**Reviewer 1**

Summary. This study investigates the response of convective storms to CCN perturbations and a warming climate, with the help of a so-called pseudo-global warming approach. The authors selected three case studies of convective storms passing over Germany and employed the ICON model in 1 km horizontal resolution with a two-moment microphysics scheme. A strong focus was put on changes in supercells, hail growth, and the underlying microphysical processes within these clouds. Overall, global warming has a dominant effect on the evolution of convective storms, with aerosol-cloud interactions only playing a secondary role. The authors highlighted several aspects important to convective storms and their evolution. However, by having this holistic approach, I believe some structured and in-depth discussions were lost. I also think the authors should try to combine more of their findings in a concise explanation and figures, which currently seem more like a checked-off list. I am detailing my major and minor comments below.

**Major comments.**

1. Case studies: I have several questions regarding the selection of the case studies. First of all, the selected days are motivated by the MOSES campaign in Germany, and often the analysed domain is focused on that part of Germany. However, no comparison to observations were done, so why this underlying motivation? If the MOSES campaign is important, then why was the third case selected? It falls out of line in several aspects throughout the analysis and it makes it difficult to fully grasp the important aspects of the findings. This is, e.g., illustrated by varying y-axes in Figure 6, 10, and 13. Is the third case really helping the analysis? Because the authors often attribute the deviations to the synoptic forcing, which to some extent is not satisfying. Moreover, it is often not clear to me what part of the domain was now analyzed for which part, which ideally should not change (i.e., Germany or MOSES) because it also makes the comparison between the results trickier. If the MOSES domain is not crucial, I would recommend that for all results the whole domain of Germany is analyzed. If the MOSES domain is crucial, then the third case study does not contribute much to the scientific findings. Also, often figures are only for one case, and the next figure shows again all cases. A more consistent approach here would be helpful. Either first doing an overview for all three and then selecting one, or show all three cases in detail. Regarding the clarity of the writing: The authors switch between the naming of the cases, i.e., Case 1-3 or the dates, and the date format changes. I would highly recommend to find a clear naming convection and stick to that. Also Figure 2 could be improved if the border of Germany and the MOSES domain are highlighted in the left columns of subfigures.

Thank you for these comments. We chose these three case studies as we were looking for convective cases with possible supercells in Germany. The two field campaigns, Swabian MOSES 2021 and Swabian MOSES 2023, were conducted by our institute, and these cases were (are) under investigation by analysing the data from the various instruments. The third case took place

during Swabian MOSES 2023 and therefore fits in with the other days. These field campaigns helped to identify candidates for our sensitivity study. We do not compare our model data to these observations because this work is a sensitivity study about aerosol effects in a warmer climate. Other cases could have been selected in principle, but these storms are currently investigated by others on the basis of observations and this work is considered to be a useful complement to that. Thus, an in-depth model evaluation is out of the scope of the present study. However, it was necessary to judge the forecast quality of the reference run. The comparison to Radar-derived precipitation amounts showed that the ICON model can reproduce the general precipitation patterns of the selected cases and therefore serves as a good basis for our sensitivity runs. To make that clearer, we included a statement in section 2.3 about that:

"No systematic comparison with observations and no evaluation in this direction are carried out, as this is not the goal of the present study. The RADOLAN data are only used to ensure that the simulations reproduce the general precipitation pattern of the selected day. The study is designed as a sensitivity study, focusing on how the model responds to changes in the prescribed parameters."

The third case was also part of these field campaigns, but the main convective activity, including supercell storms, occurred in northern Germany. Therefore, we enlarged the evaluation domain to the Germany domain. As the other two cases had storms more in southern Germany, the use of a smaller evaluation domain (the MOSES domain) makes the individual differences much clearer in contrast to averaging over a larger domain, where large parts have no convective activity. In addition, a restriction to two case studies would further weaken the validity of our results. We also tried to add information about the used evaluation domain wherever necessary.

We also restrict some of the result figures to single cases only. We do that because the remaining days show similar results. Therefore, the results can be presented more concisely without too much repetition, which improves the readability of our manuscript. Whenever results are shown for one case only, we state in the text that the other cases behave in a similar way. We do not withhold any results, but only shorten their presentation.

As suggested by the reviewer, we refer to the days now as "Case 1" etc. consistently throughout the manuscript which hopefully also improves the clarity of writing. In addition, we improved Figure 2 by using a thicker line for the German border and by including the MOSES domain rectangle in the synoptic charts in the left column as suggested.

2. Significance: Given the set of ensembles for each case study, I am missing a quantification of the uncertainty of all the investigated quantities / processes. In the text, the authors often say these changes are significant or robust without offering much evidence. While there may be systematic changes, they do not necessarily have to be significant. I would recommend to conduct a proper statistical analysis for each investigated variable. This is notable, e.g., in Figure 5, but also others, often showing only the mean over a differing areas (Germany vs. MOSES). Also in Line 354 the authors state that the hail production significantly increases with CCN, but how was this assessed? With that I disagree with the concluding remark, that these findings are robust (which they very well may be) without a proper assessment. Another question I had is the impact of the windows for the moving averages for the cold pools. Given that the storms do not live longer than 10 hours (according to Figure 3), how is a temporal window of 8 justifiable, and also where do the 166 grid points come from?

We agree with the reviewer that our wording may have been misleading. What we mostly meant was a noticeable or large increase in a quantity or a systematic behaviour. We didn't mean that in a statistical sense. When presenting our results, we removed the term "significant" completely

to avoid confusion that we might be referring to statistical significance. The text on Line 354 refers to the strong reduction of the riming process if CCNs are increased. Here, too, we simply meant a significant reduction in this process, not in a statistical sense. The text now should make this clearer.

We also adapted the text regarding the term "robust". We only use it once in the analysis of the vertical winds to state that the same behaviour was found for all three cases. The conclusion section was rewritten, and "robustness of the findings" has been deleted.

For the investigation of cold pools, we used the same method as outlined in Hirt et al. (2020). In their study, the filter size of 166 pixels corresponds to approximately 200 km horizontally. This combination of spatial and temporal (8 hr) filtering enables the identification of large cold pools while background  $\theta_{\rho}$  gradients at the coast are still sufficiently resolved to detect cold pools there. They also conducted sensitivity analyses with different spatial filtering scales and with/without temporal filtering, which did not show a strong influence on the qualitative behaviour of the results. We therefore used the same filtering criteria, although our marginally finer grid resolution results in a spatial filter of 166 km. The moving average of 8 hr is applied for every model output time step, so the entire life cycle of storms is investigated here. We have added more explanation in the method section concerning this point.

3. Storyline: The authors often talk about results and indicate "not shown". I do understand that not all results can be shown, but here I expect that a concrete storyline is built, which may also combine different figures into one, to not overwhelm the reader. However, the result sections actually starts with results that are not shown, which is not satisfying at all. Regarding the CAPE and CIN values, the authors could think of a table summarizing these numbers, or actually using the Appendix, but I would argue, that results that are not shown are definitely not the first result to be discussed. Furthermore, I believe the hail sections and their figures can be combined into one, as especially for hail the size is more important than the mass mixing ratio. Here, I actually do not understand Figure 7, as ICON with two-moment microphysics can well simulate larger hailstones than 1 cm in diameter. 1 What limits are discussed here and what curves have been extended? I do not see that in Figure 7. Restructuring the manuscript and the figures requires time, but this could be helpful to clearer convey the scientific findings. Regarding the introduction: here the sections on PGW (what exactly is perturbed in the IC and BCs mentioned in Line 29?) and convective invigoration should be extended, as in both important aspects and references are only mentioned later in the manuscript. Especially the convective invigoration is a highly debated topic, as the authors correctly state later in the manuscript, but they do not provide the proper context already in the introduction. Within the convective invigoration discussion, I would have loved to actually see changes in number concentrations of the hydrometeors, which to my surprise is not discussed at all, even though a two-moment microphysics scheme is employed. Moreover, the latent release heat could be quantified at least in terms of temperature changes within the cloud if the respective diagnostic is not available in the already produced model output. Regarding the conclusions: To me, they read more like a summary than a conclusion, as many aspects are repeated and not presented in a concise manner with clear take-home messages.

We agree with the reviewer that the result section should not start with results that are not shown. As suggested, we added a table with CAPE and CIN values and also added more text on what to expect from these indices when a PGW approach with uniform temperature modification is applied. The figures 7 and 8 were also combined. The small hail sizes are due to the averaging over larger domains. Reviewer No. 2 also had questions regarding the hail sizes, please see our reply to his/her major comment 3. In short, the hail class has an upper limit for water mass of

0.005 kg, which corresponds to hail diameters of 23 mm. Larger hailstones cannot be simulated by ICON. Maps of the dominant hail size of each grid point demonstrate that even though larger hail can be simulated, the most frequent ones do not occur at the larger end of the distribution, and the domain average will be smaller. The main statement that we want to make here is how the size distribution changes with different CCN concentrations and higher temperatures. This trend in size is described and explained, also regarding the different effects of melting due to changes in the surface-to-mass ratio. We hope that the restructuring and text modifications make it clearer to the reader.

Concerning the introduction of the PGW approach: We believe that at this point, it is sufficient to state that initial and boundary conditions are modified. Later in the section with the ICON setup and modifications, we explicitly state that besides the atmospheric temperature, soil temperature, and soil surface temperature are adjusted as well by applying the same temperature increments as imposed on the atmosphere.

We also added some text about the convection invigoration theory in the introduction as suggested: "However, this theory remains highly debated: while some studies report stronger storms under polluted conditions, others find weaker or negligible effects depending on model setup and environmental factors (Altaratz et al., 2014a; Fan et al., 2018; Igel and van den Heever, 2021). Moreover, Barthlott et al. (2022a) showed that ICON simulations do not confirm a systematic invigoration effect. This ongoing debate highlights the need to analyse aerosol impacts in parallel with thermodynamic changes."

Changes in the number concentration of the hydrometeors and latent heating are both interesting aspects. However, we believe that we already cover a large number of convection-related and microphysics-related parameters in our manuscript, and these additional sections would blow up the paper unnecessarily. Due to the extensive changes already made to the manuscript, we will refrain from these additional analyses for the time being.

The last section is called "Summary and conclusions" and therefore presents both a summary and some conclusions. We added some text and made a number of text modifications to highlight the results more clearly and also to take into account the limitations of our method.

**Minor comments.**

- 1. Line 4: the acronym ICON is missing.

  Done.
- 2. Line 5: aerosol effects on what? from context it is clouds, but it should be specified We have rephrased and clarified the text by specifying that the aerosol effects refer to their influence on the simulated clouds and precipitation of the storm events.
- 3. Line 20: "the three-day period around 23 June 2021" could be explicitly stated: Is it from 22-24? Or 21-23?
  - We have clarified the time period in the revised manuscript and now explicitly state that the hailstorms occurred from 22 to 24 June 2021.
- 4. Line 23: A reference would be nice for this event on 23 June 2021

  The reference originally cited in the preceding sentence applies to both statements. We added the reference again in the sentence for the 23 June 2021 event.
- 5. Line 86: it is not a triangular grid, but a icosahedral grid, which is trisected forming the triangles We thank the reviewer for pointing this out. We have specified the grid in a more precise way as an icosahedral-triangular Arakawa C grid.

- 6. Line 87: What is the height of the model top? This is an important information to have a better understanding of the vertical resolution.
  - We have added the information about the model top to the manuscript: "The vertical coordinates contain 100 terrain-following levels with a model top at a height of 22 km."
- 7. Line 88: SLEVE coordinates were introduced by Schär et al. 2002, so the original source should be cited Done.
- 8. Line 108: Surface temperatures are adapted accordingly in what way? Is there a formula behind it, or is the, e.g., +1 K imposed? A clearer explanation would be great.

  To avoid strong temperature gradients between the surface and the lowest atmospheric model level, the surface temperatures were adapted in the same way as the atmospheric temperatures, i.e., by applying the respective temperature increment. We have clarified this in the revised manuscript.
- 9. Line 112: I believe that 1700 cm3 CCN are rather on the high side, as also shown by Schmale et al. 2018. It is fine to use the hardcoded values in the two-moment microphysics scheme, but I believe the chosen concentrations should be better contextualized. As far as I know, these concentrations emerge from Segal and Khain, 2006, and are based on some rather early measurements of CCN.
  - We rephrased the text and included another citation to make it clearer: four different CCN concentrations are available in the Segal-Khain scheme, and the continental assumption represents typical values for central Europe and especially Germany, although very high CCN concentrations are rare in central Europe, as shown by the Schmale et al. 2018 paper. As our focus lies on aerosol-cloud interactions, we believe it is fine to use the full possible range from low to very high CCN concentrations.
- 10. Line 150: wrong chapter reference  $\rightarrow$  should be 3.2? Done. We also changed "chapter" to "section".
- 11. Line 174: "To identify ..." sentence is doubled with the next sentence of the new paragraph. To avoid redundancy, we have rephrased the second sentence. It now reads: "Helicity changes across the different temperature simulations are assessed by computing the mean values above this threshold and the number of grid cells that exceed it (Fig. 3)." This removes the duplication while keeping the intended meaning clear.
- 12. Figure 6 caption: what is meant by mm for the rain and hail? Is it a precipitation rate? The values in mm for rain and hail refer to the daily accumulated amount (24-h totals), not precipitation rate. We have clarified this in the caption of Fig. 6, which now reads: "Percentage change of the daily amount (24-h accumulated) of ... and the change in the area affected by daily totals of rain > 45 mm and hail > 10 mm ...".
- 13. Figure 8: Adding the numbers to the single tiles of the heatmap would help to grasp the figure in a faster way.
  We thank the reviewer for this suggestion. However, we decided not to add the numbers to the individual tiles of Fig. 8, as we find that the colour shading already conveys the values clearly

and adding numbers would likely reduce the readability of the figure.

14. Line 292-294: These two sentence basically say the same things, and can be combined. We agree that the two sentences partly overlap and have therefore combined them into a single, more concise sentence. The revised text now reads: "This indicates that while CC scaling explains part of the observed precipitation increase, additional dynamical and microphysical

processes must also contribute to the deviations beyond the thermodynamic expectation for cases 1 and 3, necessitating further investigation to identify the mechanisms driving these super-CC precipitation trends."

- 15. Figure 10 caption: the domainS ARE defined ... Done.
- 16. Figure 11 caption: The ending in the caption, saying the reference simulations are with C3 does not make sense to me here. All CCN variations are shown in the plots for all integrated quantities, or am I misunderstanding something?

All points represent deviations from the continental CCN assumption; therefore, the C3 deviation point is always at zero. Furthermore, all data points refer to the reference temperature scenario, as already mentioned in the text. We modified the caption text to make that clearer. Please note that in the version with tracked changes, this modification is not marked as blue due to technical reasons. The new caption reads:

"Percentage deviations of spatiotemporal averages of autoconversion (AC), accretion (ACC), evaporation (EVAP), rain freezing (RF), melting (MELT), total riming (RIM), deposition (DEP) (left column) and of total column integrated cloud water (tqc), rain (tqr), ice (tqi), snow (tqs), graupel (tqg) and hail (tqh) amounts (right column) from the respective reference run C3 with continental CCN assumption. All points refer to the reference temperature scenario without increments. The Germany domain was used, which is defined in Fig. 1"

- 17. Figure 12: Where are these vertical profiles coming from? Are these based on mean values? The hail content seems very low.
  - These vertical profiles represent averages over both time and space. As a result of this averaging, the hail content values appear relatively small. We have clarified this in the caption, which now specifies that the profiles are domain- and time-averaged.
- 18. Line 342: what is the 1 in (Fig. 11j, 1)?

  This was a typographical misunderstanding. The character is a lowercase "l," referring to subplot 11l, and not the number "1." Unfortunately, l and 1 look very similar and can easily be confused.

19. Line 364: I am wondering if the conclusion of "larger hailstones are more likely to reach the

- surface under low CCN concentrations" is circular, because under low CCN, the hailstones are larger. What is more important? So, the question is how does the melting rate change with CCN? From Figure 11, it looks more like that the melting rate is rather independent of the CCN concentration. I do not think that the sentence here is then fully correct.

  We thank the reviewer for raising this point. We agree that the conclusion in the original version was phrased too strongly and could give the impression of being circular. To address this, we have revised the text to clarify that the reduced melting efficiency of larger hailstones in low CCN environments is an interpretation based on their lower surface-to-mass ratio and on the vertical profiles of hail content. We also explicitly note that the model output only provides
  - vertical profiles of hail content. We also explicitly note that the model output only provides combined hail and graupel melting, so hail-only melting cannot be directly quantified. The revised paragraph now makes clear that the evidence for larger hailstones reaching the surface under low CCN conditions comes from the analysis of hail size distributions at the surface (Fig. 8), while the reduced efficiency of melting is presented as a consistent physical mechanism rather than a direct result of Fig. 11.
- 20. Line 368: A reference to the figure 11 should be made again from the melting rates. Done.

- 21. Line 385: The ratio of warm and cold rain and its dependence to the precipitation intensity is discussed in the next section, but the authors already mention it here, without giving much context or justification. I would move that to later.
  - This is a good suggestion, we merged the analysis of the cold and warm rain processes with the precipitation efficiency into a section now called "Rain formation processes and precipitation efficiency". We also added some more analysis text for the cold-to-warm rain processes, which was kind of short in the previous version.
- 22. Line 404: I disagree with the statement that this study investigates convective storms in Central Europe, because it does it only for Germany. This also then connects to the title, which in my opinion is misleading, as in principal Germany was looked at. The authors should consider adapting their title to that.
  - The entire simulation domain does not only cover Germany, but also parts of neighbouring countries (see Fig. 1). As we restrict the evaluation domain to Germany or the southwestern part of it, we agree with the reviewer that central Europe alone is not suitable here anymore. We changed the sentence to "in Germany". Also, the title of the paper was changed to "Aerosol effects on convective storms under pseudo-global warming conditions: insights from case studies in Germany".
- 23. Line 408: I also disagree with the statement that the chosen CCN concentrations are typical for Germany, which is not the case. These concentrations are coming from Segal and Khain, as elaborated above.
  - Please see our reply above. We study aerosol effects on clouds and precipitation and therefore use the full range of possible CCN concentrations available in the Segal-Khain scheme. What CCN concentration is typical is of minor importance here, but we provide two references stating that the continental assumption is typical for Germany. As this sentence is not essential here, we removed it and only mentioned the investigated range of different CCN concentrations in the sentence before.
- 24. Line 416: The results do not demonstrate any increase in CAPE and CIN as the authors decided not to show this. I elaborated my reasons for not doing this above.
  - We thank the reviewer for this comment. As requested, a table of CAPE and CIN values has now been added to the manuscript.
- 25. Line 448: The sentence "This intensification ..." is circular. Removing the last part "due to the heightened ..." would remedy that.
  - We agree with the reviewer that the original wording was circular and have revised the sentence accordingly. It now reads: "This intensification raises the risk of flash floods and poses a greater threat to property, agriculture, and infrastructure."

**Editorial comments.**

- 1. Line 10: wrong / incomplete latex command for the unit Done.
- 2. Line 173: plotting is colloquial and should be avoided Done.
- 3. Line 208: it is unusual to me to denote mixing ratios with r, maybe the authors can consider opting for a more common writing style
  - We agree that different notations exist for mixing ratios. To stay consistent with the definition of density potential temperature given in Hirt et al. (2020), which we explicitly cite here, we have retained the use of r to denote mixing ratios. We also note that this notation is commonly

**used in the English-speaking literature.**

**References:**

Schär, Christoph, Daniel Leuenberger, Oliver Fuhrer, Daniel Lüthi, and Claude Girard (2002). "A New Terrain- Following Vertical Coordinate Formulation for Atmospheric Prediction Models". In: Monthly Weather Review 130.10, pp. 2459–2480. ISSN: 1520-0493, 0027-0644. DOI: 10.1175/1520-0493(2002)130<2459: ANTFVC>2.0.CO;2.

Schmale, Julia, Silvia Henning, Stefano Decesari, Bas Henzing, Helmi Keskinen, Karine Sellegri, Jurgita Ovadnevaite, Mira L. Pöhlker, Joel Brito, Aikaterini Bougiatioti, Adam Kristensson, Nikos Kalivitis, Iasonas Stavroulas, Samara Carbone, Anne Jefferson, Minsu Park, Patrick Schlag, Yoko Iwamoto, Pasi Aalto, Mikko Äijälä, Nicolas Bukowiecki, Mikael Ehn, Göran Frank, Roman Fröhlich, Arnoud Frumau, Erik Herrmann, Hartmut Herrmann, Rupert Holzinger, Gerard Kos, Markku Kulmala, Nikolaos Mihalopoulos, Athanasios Nenes, Colin O'Dowd, Tuukka Petäjä, David Picard, Christopher Pöhlker, Ulrich Pöschl, Laurent Poulain, André Stephan Henry Prévôt, Erik Swietlicki, Meinrat O. Andreae, Paulo Artaxo, Alfred Wiedensohler, John Ogren, Atsushi Matsuki, Seong Soo Yum, Frank Stratmann, Urs Baltensperger, and Martin Gysel (2018). "Long-Term Cloud Condensation Nuclei Number Concentration, Particle Number Size Distribution and Chemical Composition Measurements at Regionally Representative Observatories". In: Atmospheric Chemistry and Physics 18.4, pp. 2853–2881. ISSN: 1680-7324. DOI: 10.5194/acp-18-2853-2018.

---

## Author Comment (AC2)

**Responses to the reviewers**

Pseudo-Global Warming Simulations Reveal Enhanced Supercell Intensity and Hail Growth in a Future Central European Climate

by L. Lucas et al. October 10, 2025

We thank both reviewers for reading the manuscript and providing detailed comments. We have carefully considered all comments and changed the manuscript accordingly. Please find below our responses in blue.

**Reviewer 2**

Summary: This manuscript presents an number of simulation experiments discussing supercell and hailstorm intensity under different initial conditions, by changing the surface temperature and CCN content. Three case studies are selected and experiments are performed at 4 different warming levels and 4 different CCN concentrations. The authors conclude that supercells become more intense and produce larger hail in warmer conditions. Possible reductions in CCN content due to reductions in pollution further modify these effects. While this study targets a subject that is highly relevant, it comes with many limitations and the manuscript currently overstates the conclusions that can be drawn from this. Moreover, the work is not well-situated within the broader literature. Established process changes for severe convection under climate change are not discussed properly in the context of the presented results (e.g. how increasing temperature at constant RH implies increased CAPE and CIN). While the study addresses an important topic and includes interesting case studies from a recent field campaign, I recommend major revisions before it can be considered for publication. The paper has clear potential, but currently suffers from several methodological, interpretational, and structural limitations that should be addressed to strengthen its scientific credibility and clarity. The major points are listed below, as well as line-by-line comments. These revisions are essential to ensure the study provides a robust and realistic contribution to our understanding of supercell and hail processes and their sensitivity to warming.

**Major comments.**

1. Simulation setup: While the uniform vertical warming with constant RH meets the technical definition of a PGW setup, this is a highly idealized and simplified approach. Combined with the fact, that this manuscript targets 3 case studies, no general conclusions on convective severity in a future climate should be made. Therefore, the title appears overstated with respect to the results provided. Instability changes in Europe are complex and tied to far more than uniform vertical warming. Changes in lapse rates and relative humidity are key to appropriately reflecting future instability situations in Europe. As such, the experiments performed here do not necessarily reflect a future climate state, but a generically "increased CAPE and CIN" scenario. These limitations are listed almost as an afterthought, in the last paragraph of the manuscript, which is not an appropriate location to detail essential methodological constraints.

We agree with the reviewer that no general conclusions about the future of convective storms in Germany can be drawn from these three case studies. Our primary goal was to investigate the possible effects of aerosols on clouds and precipitation in a warmer climate and to demonstrate case-to-case variability and agreement. For this purpose, we believe that the PGW simulation strategy with uniform temperature change is a valid approach for short 24-h simulation periods of convective storms. To better reflect our goal and not to overstate our results, we changed the title of the manuscript to:

"Aerosol effects on convective storms under pseudo-global warming conditions: insights from case studies in Germany"

Aerosol effects have not been studied at this high model resolution using a sophisticated double-moment microphysics scheme with an explicit hail class so far. This is the main unique aspect of our work. To better reflect that and the limitations of our simulation strategy, new text has been added in the model description, and the end of the conclusions has also been rephrased.

2. Validation of simulations: Despite picking case studies from a field campaign, no observational comparisons are conducted. Germany provides both a radar-based hail size estimate, as well as a crowd-sourced report database. Case studies from neighbouring Switzerland of the same day are not referenced, or discussed as observational reference. Precipitation data is also not used as a comparison.

We thank the reviewer for this comment. No systematic comparison with observations and no evaluation in this direction are carried out, as a model evaluation is not the goal of the present study. The RADOLAN data were only used to ensure that the simulations reproduce the general precipitation pattern of the selected day in the reference configuration. This study is designed as a sensitivity study, focusing on how the model responds to changes in the prescribed parameters.

To make this clearer, the following paragraph was added in Section 2.3 (Analysed Cases): "No systematic comparison with observations and no evaluation in this direction are carried out, as this is not the goal of the present study. The RADOLAN data are only used to ensure that the simulations reproduce the general precipitation pattern of the selected day. The study is designed as a sensitivity study, focusing on how the model responds to changes in the prescribed parameters."

We thank the reviewer for the references to the case of 28 June 2021, which have been added to the manuscript.

3. Discussion of hail size: The presented mean hail size distributions raise questions as, commonly, hail is considered to have a minimum size of 5 mm, which is the value where most distributions peak here. While Fig. 7 mostly serves to illustrate changes in distributions, commonly hail size distributions are more analyzed with respect to their larger hailstones, not just the mean. And even for the mean, these sizes appear very small. Especially for cases like June 28, 2021, where in Switzerland hail size reports exceeded 8 cm and reports, photos and sensor measurements indicate mean hail sizes well beyond 5 mm. Fig. 8 further shows dominant hailstone size, which still yields sizes well below 1 cm. Without any discussion of the observed hail sizes and a comparison to this, this poses the question, whether a) these cases are representative of damaging hail cases in southern Germany, or b) whether the modeled output sufficiently captures this. In the double-moment scheme of ICON, there exist lower and upper limits for the water mass of each hydrometeor particle class. Hail has a lower limit of 2.6E-09 kg and an upper limit of 0.005 kg. The possible hail diameters that can be simulated then range between 0.2 mm and 23 mm, larger hailstones are not possible. This also means that hail particles in ICON can be smaller than 5 mm due to these particle mass limits. We have now also added the upper limit to the text. To compute the actual hail size distribution, we enlarged the upper limit and found identical distributions, but extending to larger diameters. This procedure has already been mentioned in the text. However, the modal value of the distribution did not change. An important point to consider is the fact that Fig. 7 presents average hall size distributions over the MOSES domain: for each grid point, the size distribution was calculated based on the simulated number and mass densities; in a second step, we calculate a domain-averaged size distribution for the respective domain. We then take the modal value of this mean distribution

Figure R.1: Diameter of most frequent hail size on 23 June 2021 at 19:00 UTC.

Figure R.2: Diameter of most frequent hail size on 28 June 2021 at 16:00 UTC (left) and 19:00 UTC(right).

that, we extracted the diameter of the most frequent hailstone size (the modal value) for each grid point. The examples for 23 June 2021 (Fig. R.1) and 28 June 2021 (Fig. R.2) demonstrate the simultaneous occurrence of smaller and larger hailstones, leading to comparatively smaller values when averaged over a larger domain. In both of these examples, dominant hail sizes up to the maximum possible values are simulated. The discrepancy to the mentioned observed large hail diameters can be explained by the fact that (i) maximum hail size is limited in ICON, (ii) the tail of the observed distribution cannot be compared to the most frequent one (the modal value), and (iii) the averaging over space and time in Fig. 8. This can explain the differences in the observed hail sizes. The important point is that ICON can simulate hail formation at all, even if the maximum hail size is limited for numerical stability reasons. The discrepancy in observed hail sizes is of minor importance. The main statement that we want to make here is whether and how the size distribution changes with different CCN concentrations and higher temperatures. This trend in size is described and explained, also regarding the different effects of melting due to changes in the surface-to-mass ratio.

4. Discussion of hail size and climate change: In different parts of the manuscript, the combination of increased updraft strength and melting is discussed inconsistently. It is established, that higher-CAPE and hotter environments are expected to sort for larger hail sizes. Increased updraft strength and size increases the potential for larger hailstones. These are less affected by melting than the smaller ones. Hence, smaller stones are more likely to melt and decrease in occurrence at the surface, whereas large ones increase. Both of these aspects lead to shifts in the hail size distribution and do not contradict each other. This is not discussed coherently throughout the manuscript and not embedded in the current state of literature.

We thank the reviewer for this valuable comment. We have revised the manuscript to provide a more consistent and coherent discussion of how updraft strength and melting jointly affect hail size distributions under climate change. Specifically:

- Section 3.2.1 (Hail sizes): We clarified that stronger updrafts promote the growth of larger hailstones, which are less susceptible to melting, while simultaneously the rise in the freezing level promotes melting of smaller hailstones. This combined process results in a surface hail size distribution with fewer small stones and more large ones, consistent with Raupach et al. (2021).
- Section 3.4.1 (Microphysical processes): We added a parallel discussion showing that the same mechanism applies to changes in CCN concentration. In low CCN environments, larger hailstones form and are less affected by melting, while smaller hailstones are more likely to melt. We further noted that this mechanism is comparable to that observed in warmer environments, thereby unifying the interpretation of both temperature and aerosol effects.
- Section 4 (Conclusions): We included an explicit summary statement highlighting that stronger updrafts and melting processes act together to shift the hail size distribution toward larger stones, embedding our findings in the current state of literature Raupach et al. (2021).

We believe these changes address the reviewer's concern by making the discussion of hail size and climate change mechanisms consistent across the manuscript and better connected to existing literature.

- 5. Discussion of super-CC scaling: A brief discussion of where / how the additional precipitation falls would be very desirable in this context. Is it tied to more or longer lived convective storms? Is it tied to trailing stratiform precipitation, perhaps associated with the approaching fronts? Given that cell tracking is already performed on the data, this would provide meaningful additional information and reduce the remaining hypotheses in the manuscript.
  - We believe that we answer this question already in the manuscript, at least partly. The precipitation increase can mainly be attributed to larger CAPE values in a warmer climate, even if CIN is slightly rising as well. This has been documented in sections 3.1 and 3.2. The cell tracking revealed more numerous, but shorter-lived cells for higher temperatures, as outlined in section 3.3. Increased rain rates for higher CAPE and, as a consequence, faster depletion of CAPE were mentioned as reasons for that behaviour. The exact reasons for super-CC scaling of cases 1 and 3, however, cannot be given because of the non-linear character of microphysics and the stochastic nature of deep moist convection. We already presented possible reasons for super-CC scaling and references for them in the introduction. Now, some more text has been added in section 3.2.2 reflecting the difficulties of extracting reasons for super-CC scaling in short-term simulations of deep convection.
- 6. Discussion of INP vs CCN: The modeling experiments only touch upon the sensitivity towards CCN concentration. However, for hail formation pathways, INP concentration is a key element. E.g., hail seeding experiments aim to target INP concentration, rather than CCN concentration. At least a discussion of the conditions assumed for INPs / freezing processes should be included. We focused only on CCN effects, because recent studies showed a smaller impact of INP concentrations on total precipitation than CCN concentrations (e.g., Wellmann et al., 2020). Furthermore, the investigation of INP effects in different CCN and global warming scenarios would have drastically increased computational costs. To make clear which INP concentrations were used, we added this information in the model description part of the manuscript.

Wellmann, C., Barrett, A. I., Johnson, J. S., Kunz, M., Vogel, B., Carslaw, K. S., and Hoose, C.: Comparing the impact of environmental conditions and microphysics on the forecast uncertainty of deep convective clouds and hail, Atmos. Chem. Phys., 20, 2201–2219, https://doi.org/10.5194/acp-20-2201-2020, 2020.

7. Differentiation of supercells from all convection: The study claims that supercells in particular are intensifying and producing more hail. From my understanding, most analyses were conducted in a set model domain and not stratified by storm type. This should either be clarified, or the interpretation should be adapted. Given the importance of shear for convective organization and supercell development, the shear conditions should also be discussed somewhere. I am aware that these experiments do not systematically modify the initial wind field, however it is implied several times that the synoptic forcing plays out differently in the warming scenarios, which would, in turn, also affect the vertical shear profile and the potential for convective organization.

We thank the reviewer for this helpful comment. We have revised the manuscript to clarify the distinction between supercell-specific diagnostics and domain-wide evaluations. In the Introduction, we now explicitly state that while the case studies were chosen because they featured supercells, domain-mean analyses also capture the behaviour of other convective modes, such as single and multicells. To make this distinction clearer in the methods, we introduced a new subsection (Evaluation techniques), which emphasises that diagnostics such as cold pools, domain-mean precipitation, and convective cell numbers or lifetimes reflect the combined convective population within the model domain.

In addition, we have expanded the Results section to acknowledge the role of vertical wind shear in supercell organisation. We clarified that in the PGW framework used here, wind profiles are not systematically modified, so shear conditions remain close to the reference synoptic situations. Consequently, the changes reported for updraught helicity mainly reflect thermodynamic influences, while potential shear-related modifications are not represented in this study. To quantify wind shear and the potential for different storm types, we now include the deep-layer shear (DLS) of the reference simulation in the description of the cases. Domain averages indicate suitable conditions for supercell formation for all investigated days (case 1: 18 m/s; case 2: 20 m/s; case 3: 19 m/s).

We believe these revisions address the reviewer's concern by making it explicit which results pertain to all convection versus those most relevant to supercells, and by embedding a clear statement on the role and treatment of vertical shear.

- 8. Clarity of results: While the study follows 3 case study, single cases are highlighted at different points and it is not quite clear, why which case is focused on where. Overall, the manuscript lacks a clear storyline and presentation, seeming more like a list of results. This does not require large content changes, but rather an adaptation of the framing to make the main points more clear to the reader. Personally, I would find it helpful to have all methods in the methods section, as opposed to the beginning paragraph of each results block.
  - Most of the results are presented for all three case studies, but some results are only shown for individual ones if the other cases show similar behaviour for the sake of brevity (e.g., 2d-histograms of vertical wind in Fig. 4 only for case 2). We always mention that other cases show similar results in the manuscript. As suggested by the reviewer, we moved all technical details of our investigation to a separate part in the methods section. We believe that the readability of our manuscript is enhanced.
- 9. Missing literature discussion: A significant portion of recent severe convection literature is not

referenced or discussed in this study. A list of relevant studies is provided below, with a short note of their respective context. This contains a certain amount of Swiss literature, as the selected region is very close to the border and these studies generally extend into southern Germany, to the extent that the Swiss radar network provides coverage.

We thank the reviewer for this long list of suggestions and included most of them in our manuscript. However, we consider not all papers to be relevant for our study, and several unreviewed works have been omitted, which could be included at a later stage. Please see our individual comments below. Our reference section was already quite large and now even covers 96 entries.

**References:**

- 1. https://rmets.onlinelibrary.wiley.com/doi/full/10.1002/wea.4306 Detailed case study of June 28, 2021 in Switzerland (synoptic discussion, radar data and hail reports include southern Germany) Thanks for this very useful reference, we cited it in the description of our cases.
- 2. https://wcd.copernicus.org/articles/6/645/2025/ Ensemble model study of June 28, 2021 Has been included at the same place.
- https://arxiv.org/abs/2503.07466 (in press) Modelled current and future climatology of supercells in Europe, including hail size and environmental analysis (see heterogeneous evolution of instability in Europe)
   Has been included.
- 4. https://www.nature.com/articles/s41558-023-01852-9 Wind gust and downdraft change with climate change (thermodynamical reasoning for increasing downbursts, relevant for Fig. 4)

  Given the already extensive list of references, we do not consider this citation to be necessary.
- 5. https://egusphere.copernicus.org/preprints/2024/egusphere-2024-3924/ (accepted) Influence of Sahara dust on hailstorms in Europe
  Sahara dust acts as INP and is therefore relevant for hail formation. In our study, we do not consider changes in INP concentrations and therefore do not see a compelling reason to cite this paper.
- 6. https://journals.ametsoc.org/view/journals/mwre/152/2/MWR-D-22-0350.1.xml Topographic effects on supercells, including experiments with varying CAPE

  The suggested study investigates the effects of lakes in mountainous terrain on the evolution of supercell thunderstorms and the influence of orography on both storm intensity and occurrence frequency. As we do not focus on convection-initiating mechanisms and geographic impacts, we do not include a reference to this paper.
- 7. https://agupubs.onlinelibrary.wiley.com/doi/abs/10.1029/2024JD042828 Modeled hail climatology of Europe
  We now cite this paper in the introduction.
- 8. https://www.authorea.com/doi/full/10.22541/au.173809555.59545480 Modeled future hail climatology of Europe, including associated precipitation and environmental changes (see also moisture trends)
  - This is a pre-print which has not been peer-reviewed yet. May be included at a later date if it is accepted for publication.
- 9. https://www.nature.com/articles/s41612-023-00352-z Observational reference of severe thunderstorm types – including how many supercells have hail and vice versa Given the already extensive list of references, we do not consider this source citation to be

necessary.

10. https://egusphere.copernicus.org/preprints/2025/egusphere-2025-918/ (in press) Changing hailstorm environments with climate change

This is still a pre-print and could be included at a later date.

 $11.\ https://journals.ametsoc.org/view/journals/apme/62/11/JAMC-D-22-0195.1.xml\ Hail\ trends\ based\ on\ environmental\ changes\ (ERA-5)$

Has been added in the introduction.

- 12. https://www.researchsquare.com/article/rs-6196143/v1 Very large hail trends
  This article has not been reviewed yet, we therefore prefer not to cite it. Could be inserted at a
  later stage.
- 13. https://www.sciencedirect.com/science/article/pii/S0169809520311224 https://www.sciencedirect.com/science/sciencedirect.com/sciencedirect.com/sciencedirect.com/sciencedirect.com/sciencedirect.com/sciencedirect.com/sciencedirect.com/sciencedirect.com/sciencedirect.com/sciencedirect.com/sciencedirect.com/sciencedirect.com/sciencedirect.com/sciencedirect.com/sciencedirect.com/sciencedirect.com/sciencedirect.com/sciencedirect.com/sciencedirect.com/sciencedirect.com/sciencedirect.com/sciencedirect.com/sciencedirect.com/sciencedirect.com/sciencedirect.com/sciencedirect.com/sciencedirect.com/sciencedirect.com/sciencedirect.com/sciencedirect.com/sciencedirect.com/sciencedirect.com/sciencedirect.com/sciencedirect.com/sciencedirect.com/sciencedirect.com/sciencedirect.com/sciencedirect.com/sciencedirect.com/sciencedirect.com/sciencedirect.com/sciencedirect.com/sciencedirect.com/sciencedirect.com/sciencedirect.com/sciencedirect.com/sciencedirect.com/sciencedirect.com/sciencedirect.com/sciencedirect.com/sciencedirect.com/sciencedirect.com/sciencedirect.com/sciencedirect.com/sciencedirect.com/sciencedirect.com/sciencedirect.com/sciencedirect.com/sciencedirect.com/sciencedirect.com/sciencedirect.com/sciencedirect.com/sciencedirect.com/sciencedirect.com/sciencedirect.com/sciencedirect.com/sciencedirect.com/sciencedirect.com/sciencedirect.com/sciencedirect.com/sciencedirect.com/sciencedirect.com/sciencedirect.com/sciencedirect.com/sciencedirect.com/sciencedirect.com/sciencedirect.com/sciencedirect.com/sciencedirect.com/sciencedirect.com/sciencedirect.com/sciencedirect.com/sciencedirect.com/sciencedirect.com/sciencedirect.com/sciencedirect.com/sciencedirect.com/sciencedirect.com/sciencedirect.com/sciencedirect.com/sciencedirect.com/sciencedirect.com/sciencedirect.com/sciencedirect.com/sciencedirect.com/sciencedirect.com/sciencedirect.com/sciencedirect.com/sciencedirect.com/sciencedirect.com/sciencedirect.com/sciencedirect.com/sciencedirect.com/sciencedirect.com/sciencedirect.com/sciencedirect.com/sciencedirect.com/sciencedir

Has been added in the introduction.

14. https://www.sciencedirect.com/science/article/pii/S0169809516306020 Hailstorms and supercells in Germany

We do not consider this paper relevant to our study.

- 15. https://agupubs.onlinelibrary.wiley.com/doi/full/10.1029/2019RG000665 Review on relevant processes for hail formation and hail in a changing climate

  Has been added to the introduction.
- $16. \ https://link.springer.com/article/10.1007/s00382-024-07227-w\ Modeled\ European\ hail\ climatology$

We do not see a compelling reason to cite this paper.

17. https://amt.copernicus.org/articles/17/7143/2024/amt-17-7143-2024.html Normalization of hail size number distributions (possibly a useful reference for the conversion of microphysical output to HSD)

In the proposed study, the double-moment normalization has been used to model the shape of a series of hail size number distributions collected by a network of automatic hail sensors. We, however, compute the hail size distribution from the simulated mass and number densities directly and outline the equations used in the methods section, together with references for them. We therefore do not see a reason to include this additional reference in our manuscript.

18. https://wcd.copernicus.org/articles/2/1093/2021/wcd-2-1093-2021.html Expected changes in summer lapse rates

This reference has also been added to our manuscript.

**Line-by-line remarks.**

1. All figures: The standard color scheme used for all discrete color classes is not very color blind friendly. I would recommend to switch to something at least without red-green contrast, such as the IBM color scheme (https://lospec.com/palette-list/ibm-color-blind-safe)

We thank the reviewer for this important suggestion. In the revised figures, we have replaced the original colour scheme with the Okabe-Ito colour universal design palette, which was specifically developed to be unambiguous for people with the most common forms of colour vision deficiency. This palette avoids problematic red-green contrasts while providing eight distinct, colourblind-safe hues. The palette is described in Okabe & Ito (2008) (Colour Universal Design (CUD): How to make figures and presentations that are friendly to colorblind people, https://jfly.uni-koeln.de/color/

- 2. Line 10: format issue with unit[400]% Done.
- 3. Line 63: Acronym ICON not defined / only defined later in line 73
  We removed the mentioning of the ICON model here and included the acronym definition at the first occurrence in the abstract as well as in the remaining text.
- 4. Line 73: 1 km is not fully convection-resolving, ideally convection-permitting should be used. Done.
- 5. Line 76: acronym 2MOM is only used once, please consider replacing with double-moment Done.
- 6. Section 2.2: While the methodology is laid out in detail here, its limitations need to be discussed more thoroughly. Overall, to embed the manuscript better in the literature, it may be beneficial to include a dedicated discussion section.
  - We added some text here to address the limitations of our approach. In addition, numerous references were included throughout the manuscript as recommended by the reviewer.
- 7. Fig. 1: What was measured in Villingen-Schwenningen and how does it relate to the simulations? The reference to Villingen-Schwenningen has been removed, as measurements from this site are not directly relevant to the scope of the present study. To clarify the role of observational data in this work, a new paragraph has been added in Section 2.3 (Analysed Cases). The revised text explains that no systematic comparison with observations and no evaluation in this direction are carried out, since this is not the goal of the study. The RADOLAN data are only used to ensure that the simulations reproduce the general precipitation pattern of the selected day. This clarification also highlights that the study is designed as a sensitivity study, focusing on how the model responds to changes in the prescribed parameters. Furthermore, we improved Fig. 1b by showing fewer cities, removing the rivers, and enlarging the names of the mountains.
- 8. Fig. 2: The colorbar for precipitation should be cropped around 60

  There was a mismatch between the colorbar and the precipitation plots in the initial submission.

  This has been corrected so that the colorbar now accurately represents the plotted precipitation values. Therefore, no cropping of the colorbar is necessary anymore.
- Line 156: CIN was already defined. Please check this for all acronyms, there are multiple inconsistencies.
   Done.
- 10. Line 160: CAPE and CIN changes depend heavily on the chosen experiment, especially on changing lapse rates and RH. Maintaining lapse rates and RH basically mandates an increase in both CAPE and CIN, as the moist adiabat steepens, while everything else remains more or less constant. Moreover, the dewpoint increases less than the temperature, leading to an increase in LCL.
  - Thanks for this comment, we included more comments on that dependence in the manuscript. Moreover, a table with CAPE and CIN values has been added, as demanded by the other reviewer.
- 11. Fig. 3: and corresponding analysis: The value of 75 m2/s2 from Ashley et al., 2023, was established on 4 km resolution data. Updraft helicity is very strongly impacted by horizontal resolution. To have the same criteria, this value needs to be adjusted in relation to the resolution (this is also mentioned in the methods of Ashley et al., 2023). If the same threshold at a higher resolution is desirable, it should be justified differently.
  - The sentence in the manuscript has been revised to provide further clarification. While the

- threshold of 75 m2 s-2 in Ashley et al. (2023) was established on 4 km resolution data, the same value is retained here despite the higher resolution of 1 km, since the focus of this study is not on the exact threshold itself but on the relative changes. In addition, an analysis of the 99.95th percentile, similar to the approach of Wang et al. (2022), yielded values close to 75, which further supports the use of this threshold in the present work.
- 12. Fig. 4: I do not fully understand what is depicted in Fig. 4. What exactly is evaluated in the reference and warming simulation? And how are the percentage changes computed? Figure 4 illustrates the relative changes in updrafts and downdrafts between the reference simulation and the warming experiments. More precisely, it shows frequency differences of vertical velocity as a function of height and wind magnitude for different warming scenarios with respect to the reference temperature evaluated over the MOSES domain and the full 24-h simulation period. We have adapted the Figure caption to make it clearer. Due to technical reasons, this modification does not show up in the version with tracked changes. Positive values indicate an increase under warming, and negative values indicate a decrease.
- 13. Line 199: Why is the CCN sensitivity not shown? At least this should be included in an Appendix or Supplement. The following description of the vertical structure is not very clear without any supporting material.

  We have now included the CCN sensitivity plots in the main text (new Fig. 5), which provide supporting material for the description of the vertical structure.
- 14. Line 204-214: All descriptions of methodology / computations should be contained in the methods section.
  - We thank the reviewer for this comment. All methodological descriptions have now been moved to the Methods section to ensure consistency in the manuscript structure.
- 15. Line 228: Considering that the experiment does not include changes to the large-scale dynamics, could you please elaborate how the synoptic forcing is expected to change?

  The text has been revised to clarify that the imposed warming experiments do not modify the large-scale dynamics, and that the prevailing synoptic environment of each analysed day remains unchanged. In addition, a short discussion was added on how synoptic forcing is expected to change in a warmer climate (e.g., poleward shifts of storm tracks and jets, and changes in cyclone and frontal characteristics; Harvey et al., 2020; Priestley and Catto, 2021).
- 16. Fig. 6: Given the constraint on xmax for the derived hail size distribution, how does this affect the possibility to meaningfully calculate the 95th percentile of the HSD?

  There seems to have been a misunderstanding: Fig. 6 does not show the hail size distribution but rather the daily amounts (24-h accumulated) of precipitation in mm. To avoid confusion, the caption of Fig. 6 was revised accordingly and now explicitly states that daily totals of rain and hail are analysed, including the 95th percentile of these accumulated amounts.
- 17. Lines 245-257: Assuming the same method is used to obtain Fig. 6, this methodological explanation should be earlier, ideally in the methods section.

  As clarified above, Fig. 6 does not present hail size distributions but rather daily accumulated precipitation amounts (rain and hail in mm). As these quantities simply represent daily accumulated precipitation amounts, we believe that no additional methodological explanation is needed, and therefore, a revision of the methods section is not required.
- 18. Line 261: Enhanced melting does not immediately mean shifting the distribution peak towards smaller sizes, as it affects small stones more than large stones and can skew the HSD towards the tail. Moreover, hotter conditions are often related to greater instability and greater updraft

speeds, especially in this simulation setup, where RH is kept constant and higher temperatures basically just increase CAPE.

We thank the reviewer for this helpful clarification. We agree that the original formulation was misleading, as it implied that enhanced melting necessarily shifts the hail size distribution peak toward smaller sizes. In the revised manuscript, we have removed this statement and instead clarified that a rise in the freezing level primarily enhances the melting of smaller hailstones, while larger hailstones produced in stronger updrafts are less affected. This distinction avoids the previous confusion and is now consistent with the subsequent discussion of hail size distributions and with the current state of the literature (e.g., Raupach et al., 2021).

- 19. Line 283: Is a vapor content increase close to CC in line with climate projections for central Europe? While moisture trends have much greater uncertainty than temperature trends, a number of models suggest a drying throughout much of Europe, including southern Germany (e.g. Fig. 4 in Thurnherr et al., 2025). This needs to be discussed somewhere, as a constant RH can not immediately be presumed to be representative of a future climate state.
  - The mentioned Thurnherr paper seems not to be published yet; we only found a preprint which is not citable. However, we found a new publication by Feldmann et al. (2025) that apparently uses the same data set. They compare a current climate simulation with a pseudo–global warming +3°C global warming scenario and find that the future climate simulation shows an average increase of supercell occurrence by 11%. However, there is a spatial dipole of change with strong increases in supercell frequencies in central and eastern Europe and a decrease in frequency over the Iberian Peninsula and southwestern France. In central Europe, 2-m specific humidity increases, so that 2-m relative humidity only slightly decreases, which would be similar to our approach with constant relative humidity. We like to point out that our focus lies on case studies on aerosol–cloud interactions with sophisticated microphysics, including a separate hail class for a detailed process understanding. We included this reference and some remarks on why we cannot study long-term impacts on humidity at the end of section 3.2.
- 20. Line 289ff: Can you visualize the precipitation associated with the front and with convection to support your hypothesis?

We changed the sentence from "a pronounced synoptic front" to "in the presence of moderate to strong large-scale synoptic ascent supporting convection initiation (see Fig. 2e)."

It is not possible to separate the precipitation associated with the front and with convection as they are connected. The precipitation distribution resembles more a squall line than stratiform frontal precipitation and agrees quite well with Radar observations (Fig. R.3).

Figure R.3: 30-min precipitation rate from ICON reference run (a) and radar-derived 1-h precipitation amount (b).

21. Section 3.3: The tracking methodology should be detailed in the methods. How are splits and mergers handled, how are they "counted" for the cell number? Tracking algorithms can be quite sensitive to this and hence also the total cell number. The mentioned behavior of faster convective lifecycles may apply to multicell and single-cell thunderstorms, but should not apply to supercell thunderstorms (the initial main focus of this study), as they normally propagate along long tracks, as long as the encountered environment remains favorable. While the listed explanation may apply to less severe storms occurring in the model domain, this differentiation should be made clear.

The description of the Tobac tracking methodology has now been moved to the Methods section, and additional detail has been added regarding how splits and mergers are handled. Specifically, Tobac links features between consecutive time steps using a nearest-neighbour and overlap-based approach. When a feature splits into multiple successors, the trajectory continues along the largest successor while the others are recorded as new tracks. In the case of mergers, the trajectory of the largest predecessor is continued, and the additional features are terminated. In this study, the number of convective cells, therefore, refers to the number of unique tracks initiated during the simulation period.

We agree with the reviewer that supercells usually propagate along long tracks, as long as the encountered environment remains favorable. In our tracking of convective cells with Tobac, we do not differentiate between different types of convection; all cells are tracked, independent of their lifetime or track length. By doing this, we find on average a reduction in the lifetime of cells. We also restricted the computation of the averages to lifetimes greater than 120 minutes and found the same systematic behaviour. So our statement that convective storms on average have shorter lifetimes is valid. Furthermore, the formation of supercells can be strongly dependent on the aerosol load, as different CCN concentrations can determine whether a supercell is successfully simulated or not (Barthlott et al., 2024).

We modified our text to make it clear that this finding is valid for all detected convective cells and not specifically for supercell storms.

- 22. Line 361: Here the smaller sensitivity of larger hail to melting is mentioned, but this should be consistently discussed throughout the manuscript sections.

  Such a statement is now written at all suitable places, i.e., in section 3.2.1, 3.4.1, and in the summary.
- 23. Line 378: Could the relevance of the synoptic forcing be verified by providing a spatial visual-ization of the cold-to-warm ratio? The presence of the forcing is mentioned in many hypotheses throughout the manuscript, without attempt to verify them.
  We thank the reviewer for this helpful comment. The reference to synoptic forcing in this context was based only on an assumption and may not be directly related to the frontal system. To avoid misinterpretation, this statement has been removed from the manuscript. We particularly like the idea of a spatial visualization of the cold-to-warm ratio, which will be pursued in future work.
- 24. Line 386: Is there a specific reason to choose the unit mm / 30 min?

  Yes. The model output is available at 30-minute intervals. Consequently, precipitation amounts are reported in mm per 30 minutes, which directly reflects the model's temporal resolution.
- 25. Section 3.4.3: It is a bit unclear to me, which point exactly is being made here. If precipitation efficiency increases, but generation decreases and hence total precipitation decreases, why are not all 3 terms discussed in combination here? Why does precipitation efficiency require a separate analysis?

Section 3.4.3 has been revised to clarify that precipitation efficiency (PE) is not an independent measure but a diagnostic linking total precipitation (P) and hydrometeor generation (G). We also merged this section with section 3.4.2 about the cold and warm rain processes, as the same processes are analysed there. This was also a suggestion from the other reviewer. The combined text now explains that PE is discussed separately because it provides insight into how efficiently generated condensate is converted into surface precipitation, highlighting microphysical processes such as evaporation and melting. We also make explicit that the decrease in precipitation under higher CCN concentrations results from the combined effect of decreasing G and increasing PE.

- 26. Line 405: Consider replacing convection-resolving with km-scale or convection-permitting Done.
- 27. Line 416: Increasing temperatures vertically homogeneously, while maintaining constant RH, basically mandates an increase in CIN and CAPE. This is inherent to the simulation setup and not a finding. The CAPE and CIN conditions, as well as their changes are not shown in detail in the first place, results that are not shown should not constitute a main point of the conclusion. A table containing CAPE and CIN values has been added to the manuscript to provide a clearer basis for this discussion. While the general increase in CAPE and CIN is indeed inherent to the PGW setup with vertically homogeneous warming and constant RH, the results also show case-to-case variability and nonlinear interactions with other processes. For this reason, we included these analyses to test and illustrate how such interactions manifest in the simulations.
- 28. Line 418ff: Changes like this are highly regionally dependent and should not be generalized to this extent. What is exactly is meant by stronger storms? Which variable?

  To avoid overgeneralization and to be more precise, the sentence has been revised. The new formulation emphasises that simultaneous increases in CAPE and CIN imply the need for stronger triggering mechanisms to initiate convection, which in turn favours the development of fewer but more intense convective systems, rather than broadly referring to "stronger storms."
- 29. Line 421ff: Supercell frequency and intensity are two different matters and should be discussed separately. Frequency overall cannot be addressed by case studies, as the frequency of supercellfavorable days cannot be deduced from this. The simulations have the data necessary to identify whether UH is increasing in intensity per storm, increasing its area per storm, or if there are more storms. Ideally this should be addressed quantitatively and not left to speculation. We agree with the reviewer that no trends regarding the future occurrence of supercells can be derived from our work as the number of days with suitable atmospheric conditions can not be assessed. Simulations in climate mode are necessary for that, e.g., the study of Feldmann et al. (2025), which is also cited in our manuscript. The question about more storms has been answered by tracking each convective cell with the Tobac tracking tool in section 2.4.3. However, we did not differentiate between different types of convection and included averages of the number of detected cells and the mean lifetime over all tracked elements. Focusing on longer-lived storms did not change the characteristic dependence; please also see our reply to the comment on Section 3.3. We present only mean values of helicity and state that the increase in the mean value can be caused by more supercells, more intense storms, or a combination of both. We believe that this statement is valid in its current form. We also find increased rain intensities in the warming scenarios, which supports this statement. To assess the UH of each individual storm is a difficult task and would require selecting the thermodynamic environment of each storm during the entire lifecycle. Since our focus is more on the interactions between aerosols and clouds in a warmer climate, this comprehensive work would go beyond the scope of the present study. We therefore limit ourselves to the previous statement that the UH increase

can be caused by more supercells, more intense storms, or by a combination of both.

30. Line 447: This should be rephrased to reflect the limitations of the PGW setup and the nature of case studies.

We rephrased the end of the conclusions to better reflect the limitations of our simulation strategy.

---

## Referee Report (RR1)

**Review for "Aerosol effects on convective storms under pseudo-global warming conditions: insights from case studies in Germany" by Lucas et al.**

Thank you to the authors for working through the extensive comments of both reviewers. I believe the manuscript improved greatly. For future review processes, I recommend to the authors to add the line numbers of the changes (from the final, revised manuscript) in the responses to the reviewers. This way, it is easier to see where the changes were made and how they now fit in the adapted manuscript. I refer to the line numbers of the final manuscript here.

**Major comments**

- **Domain averages:** I agree with the authors that domain-averages smear out the signal, however, for convective storms it makes sense to first define a threshold, e.g., via total hydrometeor mass and then average, such that grid points with no storm are excluded. This should be considered when conducting domain averages (e.g., Figure 12).

- **Heatmaps:** The authors argued that additional numbers in the heatmaps (such as Figure 8) are not needed, which I still disagree with, however, at a minimum the same range for should be used such that a direct comparison across the three cases is possible. I would still say numbers should be add as well.

**Minor comments**

- Line 2: emissionS regulations $\rightarrow$ emission regulations

- Line 100: The model description is missing the model time step as well as the output frequency.

- Line 105: What are summertime INP concentration? Please add the number concentration, as it is kept constant.

- Line 134: The authors responded that CCN concentrations of $1700\,\mathrm{cm}^{-3}$ are rare in the response to reviewer 1, but in the main text they chose to still state that these are typical conditions. Please remedy that and just define C3 as the reference, which is also fine.

- Line 215: What density is assumed for the hail particles?

- Line 380: References missing

---

## Referee Report (RR2)

Review 2 of **"Aerosol effects on convective storms under pseudo-global warming conditions: insights from case studies in Germany"**

The authors undertook a great effort to implement all comments. The manuscript is much improved, especially in terms of framing the findings and improving readability. A few points remain, they are detailed in the following:

1. Figs 3 and 4: part of my confusion stemmed from the fact that the colorbar extends to -300% frequency change, which should not occur as a relative change. Looking at the values in the Figures, it appears they actually never decrease under -100%. The colorbar should be cropped to avoid confusion.

2. Hail size discussion and observational references: While I understand that an absolute reproduction of observations is not the goal here, at least a literature-based discussion of absolute values is warranted. Figs R1 and R2 show that the max. hailsize does indeed appear realistic and I think it is important for the reader to get this impression from the manuscript. These figures could e.g. be added in a supplement. The same goes for observational comparisons from radar data or the field campaign. If anything, it is an opportunity to increase the credibility of the results, without claiming a verification of the simulation.

3. Supercells vs all storms and updraft helicity: I don't quite follow, why the changes in UH cannot be split into changes in storm number vs changes in UH intensity. A cell tracking has already been performed, so the number of cells meeting supercell criteria should be easily identifiable, as well as their respective mean UH values / UH areas. Sure, the statement on UH changes can be attributed to both cell number and intensity changes is valid, but it would be nice to explicitly state this here, given that the data is available. I am aware that the revised version focuses more on convection overall and has a less pronounced supercell focus.

4. Constant RH: Thurnherr et al. 2025 does show a decrease in RH of ~3% for central Europe in the summer months, when the case studies take place. But this discussion can hinge on any number of climate models producing RH trends for central Europe. WCD - A pan-European analysis of large-scale drivers of severe convective outbreaks also shows a decrease of 1-3% per decade based on ERA5 trends. I don't want to convince you to cite this paper, but just point out that this decision warrants a justification or short discussion.

Minor remarks:

- Both The Effect of 3° $C Global Warming on Hail Over Europe - Thurnherr - 2025 - Geophysical Research Letters - Wiley Online Library and NHESS - Insights from hailstorm track analysis in European climate change simulations were published at the time when the revised manuscript was submitted.

- Line 519: Feldmann et al. 2025 actually show no significant changes in updraft velocity. The way it is phrased currently, could be misleading what exactly this is referring to.
- There are still some instances of convection-resolving instead of convection-permitting.

Given the extensive amount of literature recommended that also included my own work, I waive my anonymity at this point.

With best regards,

Monika Feldmann

---

## Author Response (AR2)

**Responses to the reviewers**

Aerosol effects on convective storms under pseudo-global warming conditions: insights from case studies in Germany

by L. Lucas et al.                                                               December 4, 2025
* * *
We thank both reviewers and the editor for reading the manuscript again and providing detailed comments. We have carefully considered all comments and changed the manuscript accordingly. Please find below our responses in blue.

**Comments from the editor**

1. There is some jumping around in the abstract between responses to warming and CCN. For example, the sentence "In some cases, ... thermodynamic expectations." could be moved higher with the other warming-related content.
   Thanks for this remark, we moved that sentence higher as suggested so that the warming-related content is together.

2. The caption of 8a) says "...using the continental CCN (C3) for case 1." Is this correct? Is C3 only used in case 1, or in all cases?
   Thanks for pointing that out, the caption text for Fig. 8a was misleading with respect to the evaluation domains. Figure 8a shows only size distributions for continental CCN for case 1 and illustrates the response to different warming scenarios. Figs. 8b-d, however, show the results of all 4 CCN concentrations for all three cases. We rephrased the caption to make that clear.

3. It would be nice to have uniform colormap limits for panels b, c, and d of Figure 8.
   Good point, we now use the same colormap boundaries for these figures.

**Comments from Reviewer 1**

Thank you to the authors for working through the extensive comments of both reviewers. I believe the manuscript improved greatly. For future review processes, I recommend to the authors to add the line numbers of the changes (from the final, revised manuscript) in the responses to the reviewers. This way, it is easier to see where the changes were made and how they now fit in the adapted manuscript. I refer to the line numbers of the final manuscript here.

We thank the reviewer for reading our manuscript again and providing additional feedback. As we uploaded a version with tracked changes, we did not include line numbers in our responses, but will do so in the future.

**Major comments**

- Domain averages: I agree with the authors that domain-averages smear out the signal, however, for convective storms it makes sense to first define a threshold, e.g., via total hydrometeor mass and then average, such that grid points with no storm are excluded. This should be considered when conducting domain averages (e.g., Figure 12).
  We agree with the reviewer that adding a threshold to analyse cloudy grid points only would lead to a more distinct signal. However, if the number of clouds is different, the signal could lead to wrong conclusions as for example few strong convective clouds could dominate the signal. To be able to really compare, if more or less hail is simulated in a specific evaluation volume, the domain must be identical for all different sensitivity studies. Especially Fig. 12 with domain-averaged hail profiles could look completely different if only cloudy grid points had been used

for averaging. We therefore believe that the technique of using the entire evaluation domain is adequate.

- Heatmaps: The authors argued that additional numbers in the heatmaps (such as Figure 8) are not needed, which I still disagree with, however, at a minimum the same range for should be used such that a direct comparison across the three cases is possible. I would still say numbers should be add as well.
  We decided to follow the reviewer's suggestion and added the numbers in these plots. Moreover, we now use the same colormap range in all subplots.

**Minor comments**

- Line 2: emissionS regulations → emission regulations
  Done

- Line 100: The model description is missing the model time step as well as the output frequency.
  We added this sentence in L111-113 of the revised manuscript:
  *"A time step of 10 s is used, and the data is written out every 30 min. In addition, data for tracking convective cells is written out every 5 min."*

- Line 105: What are summertime INP concentration? Please add the number concentration, as it is kept constant.
  The INP concentration (immersion and deposition) depend on the temperature and relative humidity over ice, so there is not one value that we can give here. An equation for calculating the INP concentration together with profiles over Germany is given in Hande et al. (2015). This paper is already cited in the text where we state that summertime conditions are used. As the INP concentration remains constant, we believe that the reference to the Hande paper is sufficient.

- Line 134: The authors responded that CCN concentrations of 1700 cm$^{-3}$ are rare in the response to reviewer 1, but in the main text they chose to still state that these are typical conditions. Please remedy that and just define C3 as the reference, which is also fine.
  In our reply to reviewer 1, we stated that the continental assumption represents typical values for central Europe and especially Germany, although very high CCN concentrations are rare in central Europe, as shown by the Schmale et al. 2018 paper. We refer to continental CCN concentrations as "high" and continental polluted as "very high", this is probably the misunderstanding here. In the manuscript, we clearly state that C3 (continental) is the reference: L138-139: *"The continental CCN concentration (C3) is chosen as the reference concentration, as this aerosol assumption represents typical conditions of central Europe (Hande et al., 2016; Costa-Surós et al., 2020)"*

- Line 215: What density is assumed for the hail particles?
  The density of pure ice used in ICON is 916.7 kg/m$^3$.

- Line 380: References missing
  Thanks for finding that mistake, we included the correct reference now.

**Comments from Reviewer 2**

The authors undertook a great effort to implement all comments. The manuscript is much improved, especially in terms of framing the findings and improving readability. A few points remain, they are detailed in the following:

We thank the reviewer for reading our manuscript again and providing additional feedback.

**Major comments**

1. Figs 3 and 4: part of my confusion stemmed from the fact that the colorbar extends to -300% frequency change, which should not occur as a relative change. Looking at the values in the Figures, it appears they actually never decrease under -100%. The colorbar should be cropped to avoid confusion.

   We believe that the reviewer refers to Figs. 4 and 5 with the 2d-histograms of vertical velocity. We agree with the reviewer that reductions by -300% do not occur, it was just our goal to have the white colors at zero. With this pre-defined colormap, the boundaries have to be between +300 and -300%. We now adapted the colorbar to show only the meaningful range between -100% and +300%.

2. Hail size discussion and observational references: While I understand that an absolute reproduction of observations is not the goal here, at least a literature-based discussion of absolute values is warranted. Figs R1 and R2 show that the max. hailsize does indeed appear realistic and I think it is important for the reader to get this impression from the manuscript. These figures could e.g. be added in a supplement. The same goes for observational comparisons from radar data or the field campaign. If anything, it is an opportunity to increase the credibility of the results, without claiming a verification of the simulation.

   We disagree with the reviewer at this point. Our main objective was to see how the hail size distribution changes with different CCN concentrations and higher temperatures. Comparisons to observed maximum hail sizes (which, by the way, are not available for all investigated cases here) is difficult as the upper end of the size distribution is limited for numerical stability reasons. We therefore decided just to investigate how the simulated most dominant hail sizes react to CCN and temperature modifications. Figs. R1 and R2 also did not show the maximum hail size, but the dominant one (the diameter where the size distribution has its peak). Including these figures in the paper would not really help the reader; therefore, we decided to add an additional statement in the text (L354-357) to make clear that the small values are an effect of averaging:

   *"Because domain averages of the dominant hailstone sizes are computed from individual size distributions at each grid point, the mean values in Fig. 8b–d are comparatively small and always below 1 cm in diameter. It should be pointed out that larger dominant values up to the maximum extent of the hail size distribution do occur when their spatial distribution in the respective evaluation area is analysed (not shown)."*

   As this work is not intended as a model evaluation, we believe that an intercomparison with Radar data or specific campaign data is not necessary. This would be the topic of an entirely new paper and does not fit into this manuscript anymore. However, we have compared the reference setup with radar data to ensure that the model reproduces the general precipitation patterns of the selected day, which we believe is an important basis for our sensitivity runs. This has been mentioned in section 2.3 about the analyzed cases. We do hope that our point of view is convincing.

3. Supercells vs all storms and updraft helicity: I don't quite follow, why the changes in UH cannot

be split into changes in storm number vs changes in UH intensity. A cell tracking has already been performed, so the number of cells meeting supercell criteria should be easily identifiable, as well as their respective mean UH values / UH areas. Sure, the statement on UH changes can be attributed to both cell number and intensity changes is valid, but it would be nice to explicitly state this here, given that the data is available. I am aware that the revised version focuses more on convection overall and has a less pronounced supercell focus.

Thank you for the comment and the opportunity to clarify this point. A cell-tracking algorithm was indeed applied in our analysis; however, it identifies all convective cell types, not only supercells. While this allows us to quantify the total number of detected convective cells, it does not directly separate supercells from other convective cells in a way that would allow us to robustly track changes in their UH characteristics alone.

Even if we were to extract the subset of tracked cells that meet the supercell criteria, an additional level of analysis would be required to disentangle whether changes in UH arise from (1) an increase in the number of supercells, or (2) changes in their structure, such as larger UH areas or more intense UH maxima.

Because UH depends not only on storm intensity but also on the spatial extent of rotating updrafts, answering this question rigorously would require computing and analysing cell sizes and UH-area metrics, which are not part of the existing tracking output. This goes beyond the scope of our current analysis, which in the revised version focuses more broadly on convection rather than on supercells specifically.

We therefore maintain that UH changes can arise from both storm-number and storm-intensity effects, but explicitly separating these contributions would require a dedicated analysis not included here.

4. Constant RH: Thurnherr et al. 2025 does show a decrease in RH of 3% for central Europe in the summer months, when the case studies take place. But this discussion can hinge on any number of climate models producing RH trends for central Europe. WCD - A pan-European analysis of large-scale drivers of severe convective outbreaks also shows a decrease of 1-3% per decade based on ERA5 trends. I don't want to convince you to cite this paper, but just point out that this decision warrants a justification or short discussion.

Thank you for pointing this out. We have added a clarification in the Methods section explaining our choice to assume constant relative humidity in the PGW perturbations. Although several studies show decreasing summer RH trends over Central Europe, our PGW setup follows the standard approach in which specific humidity is recomputed under constant RH. This choice is intentional, as it isolates the thermodynamic effect of uniform warming. Since the aim of our study is to assess the sensitivity of convective storms to temperature perturbations and CCN concentrations, allowing RH to vary would introduce additional changes in moisture distribution and confound the interpretation. The new text has been added to Section 2.2. (L122-127)

**Minor comments**

1. Both The Effect of 3° C Global Warming on Hail Over Europe - Thurnherr - 2025 - Geophysical Research Letters - Wiley Online Library and NHESS - Insights from hailstorm track analysis in European climate change simulations were published at the time when the revised manuscript was submitted.
Thanks for pointing that out, we included both references in the introduction.

2. Line 519: Feldmann et al. 2025 actually show no significant changes in updraft velocity. The way it is phrased currently, could be misleading what exactly this is referring to.

This was meant with respect to the likelihood of supercells. We rephrased the text (L530-533), it now reads: *"The increased likelihood of supercells found by our single case studies agrees well with recent climate simulations of Feldmann et al. (2025). By comparing a current-climate simulation with a pseudo–global warming scenario (+3 K), they found that the future climate simulation shows an average increase of supercell occurrence by 11%"*

3. There are still some instances of convection-resolving instead of convection permitting.
   Done